# Soil moisture estimates at 1-km resolution making a synergistic use of Sentinel data

Remi Madelon[1], Nemesio J. Rodríguez-Fernández[1], Hassan Bazzi[2], Nicolas Baghdadi[2], Clement Albergel[3], Wouter Dorigo[4], and Mehrez Zribi[1]

[1]CESBIO (Université de Toulouse, CNES, CNRS, IRD, INRAE), 18 av. Edouard Belin, bpi 2801, 31401 Toulouse, France
[2]TETIS, INRAE, Université de Montpellier, 34090 Montpellier, France
[3]European Space Agency Climate Office, ECSAT, Harwell Campus, Oxfordshire, Didcot OX11 0FD, United Kingdom
[4]TU Wien, Vienna, Austria

**Correspondence:** Nemesio J. Rodríguez-Fernández (nemesio.rodriguez@cesbio.cnes.fr)

**Abstract.** Very high-resolution ($\sim 10 - 100$ m) surface soil moisture (SM) observations are important for applications in agriculture, among other purposes. This is the original goal of the the $S^2MP$ (Sentinel-1/Sentinel-2 derived Soil Moisture Product) algorithm, which was designed to retrieve surface SM at agricultural plot scale using simultaneously Sentinel-1 (S1) backscatter coefficients and Sentinel-2 (S2) NDVI (Normalized Difference Vegetation Index) as inputs to a neural network trained with Water Cloud Model simulations. However, for many applications, including hydrology and climate impact assessment at regional level, large maps with a high-resolution (HR) of around 1 km are already a significant improvement with respect to most of the publicly available SM datasets, which have resolutions of about 25 km.

In this study, the $S^2MP$ algorithm was adapted to work at 1-km resolution and extended from croplands to herbaceous vegetation types. A target resolution of 1 km also allows to evaluate the interest of using NDVI derived from Sentinel-3 (S3) instead of S2. Two sets of SM maps at 1-km resolution were produced with $S^2MP$ over 6 regions of $\sim 10^4$ km$^2$ in the southwest and southeast of France, Spain, Tunisia, North America, and Australia for the whole year of 2019. The first set was derived from the combination of S1 and S2 data (S1+S2 maps), while the second one was derived from the combination of S1 and S3 (S1+S3 maps). S1+S2 and S1+S3 SM maps were compared to each other, to those of the 1-km resolution Copernicus Global Land Service (CGLS) SM and Soil Water Index (SWI) datasets, and to those of the SMAP+S1 product.

The $S^2MP$ S1+S2 and S1+S3 SM maps are in very good agreement in terms of correlation ($R \geq 0.9$), bias ($\leq 0.04$ m$^3$ m$^{-3}$) and standard deviation of the difference ($STDD \leq 0.03$ m$^3$ m$^{-3}$) over the 6 domains investigated in this study. In a second step, the S1+S3 $S^2MP$ maps were compared to the other HR maps. S1+S3 SM maps are well correlated to the CGLS SM maps ($R \sim 0.7$-$0.8$), but the correlations with respect to the other HR maps (CGLS SWI and SMAP+S1) drop significantly over many areas of the 6 domains investigated in this study. The highest correlations between the HR maps were found over croplands and when the 1-km pixels have a very homogeneous land cover. The bias in between the different maps was found to be significant over some areas of the six domains, reaching values of $\pm$ 0.1 m$^3$ m$^{-3}$. The S1+S3 maps show a lower STDD with respect to CGLS maps ($\leq 0.06$ m$^3$ m$^{-3}$) than with respect to the SMAP+S1 maps ($\leq 0.1$ m$^3$ m$^{-3}$) for all the 6 domains.

Finally, all the HR datasets (S1+S2, S1+S3, CGLS and SMAP+S1) were also compared to in-situ measurements from 5 networks across 5 countries along with coarse resolution (CR) SM products from SMAP, SMOS and the ESA Climate Change

Initiative (CCI). While all the CR and HR products show different bias and STDD, the HR products show lower correlations than the CR ones with respect to in-situ measurements. The discrepancies in between the different HR datasets except for the more simple land cover conditions (homogeneous pixels with croplands) and the lower performances with respect to in-situ measurement than coarse resolution datasets, show the remaining challenges for large scale HR SM mapping.

## 1 Introduction

Surface soil moisture (SM) plays a key role in the Earth water cycle as it affects many hydrological processes such as infiltration, runoff, evaporation and precipitation (Koster et al., 2004). SM measurements are used to constrain numerical weather prediction (NWP) models via data assimilation (De Rosnay et al., 2013; de Rosnay et al., 2014; Rodríguez-Fernández et al., 2019) as well as for crop yields forecasting, food security and agriculture management (Guerif and Duke, 2000). SM was identified as one of the 50 "essential climate variables" (ECVs) by the Global Climate Observing System in the context of the United Nations

Framework Convention on Climate Change (Plummer et al., 2017; GCOS, 2021). Building long time series of SM is crucial for climate applications, and this is the goal of projects such as the European Space Agency's Climate Change Initiative (ESA CCI) for SM (Gruber et al., 2019).

Both active and passive microwave sensors can be used to estimate SM at coarse resolutions ($\sim$ 25-40 km) including the active Advanced Scatterometer (ASCAT, Vreugdenhil et al., 2016), the passive Advanced Microwave Scanning Radiometer 2

(AMSR2, Kim et al., 2015; Imaoka et al., 2000) and the two sensors that have been specifically designed to measure SM at L-band: the Soil Moisture and Ocean Salinity (SMOS, Kerr et al., 2012) and Soil Moisture Active Passive (SMAP, Entekhabi et al., 2010). However, despite the actual availability of these SM products, they do not match the requirements of a number of applications. Peng et al. (2020) have discussed a road-map and requirements for future SM products. An optimal spatial resolution for data assimilation into NWP models and reanalysis would be 5-10 km (global models are already running with

resolutions better than 10 km, see for instance Muñoz-Sabater et al., 2021). The evaluation of climate models and the assessment of climate change impacts at regional level would also benefit from a higher resolution than that of the current generation of coarse resolution sensors. In addition, other applications in hydrology, agriculture and risk assessment require even higher resolutions of $\sim$ 1 km (Massari et al., 2021).

Downscaling the coarse scale resolution data by merging them with higher resolution data is a possibility to achieve high

resolution SM datasets. For example, high resolution SM estimates can be derived from visible/infrared (Merlin et al., 2012) or Synthetic Aperture Radar (SAR, Tomer et al., 2016; Das et al., 2019) measurements. SAR observations alone have also been tested to estimate SM using different frequencies and instruments such as RadarSAT, ALOS-L or TerraSAR-X. Radar signal is sensitive to the dielectric constant linked to SM but also to surface geometry (including roughness) and vegetation water content and structure (Ulaby et al., 1986). Different inversion algorithms have been proposed considering principally

three techniques: change detection algorithms (Wagner et al., 1999; Balenzano et al., 2010; Bauer-Marschallinger et al., 2018), direct inversion of physical or empirical models (Moran et al., 2000; Srivastava et al., 2009; Pierdicca et al., 2010; Hajj et al.,

2014; Bousbih et al., 2017; Şekertekin et al., 2018) and finally machine learning methods (Paloscia et al., 2004; Notarnicola et al., 2008; El Hajj et al., 2017).

With the successive launches of the C-band SARs onboard Sentinel-1 A (S1A, 2014) and Sentinel-1 B (S1B, 2016), SM can be estimated at high spatial resolution and with a revisit time better than 6 days over Europe. Three operational high resolution (HR) SM datasets at 1-km resolution using S1 exist such as the S1 SM and Soil Water Index (SWI) products from the Copernicus Global Land Service (CGLS, Bauer-Marschallinger et al., 2018, 2019) and the SMAP+S1 downscaled product (Das et al., 2020). SM estimates at a very high resolution (10-m scale) at some locations in Europe (as well as in Lebannon and Morocco) over croplands are also distributed by the French continental surfaces data center (THEIA, https://www.theia-land.fr). In contrast to the CGLS datasets, the THEIA SM dataset is obtained using synergistically S1 and Sentinel-2 (S2) measurements as inputs to the Sentinel-1/Sentinel-2 derived Soil Moisture Product ($S^2MP$) algorithm (El Hajj et al., 2017). This dataset has been evaluated against in-situ measurements in comparison to SMAP, SMOS and ASCAT coarse resolution (CR) datasets (El Hajj et al., 2018) and with respect to the CGLS S1 SM dataset (Bazzi et al., 2019), both in the south of France. The $S^2MP$ SM estimates showed the lowest unbiased root mean squared errors with respect to in-situ measurements but also a moderate correlation, lower than that obtained for SMAP and ASCAT datasets (El Hajj et al., 2018). In this region, the $S^2MP$ SM showed better performances with respect to in-situ measurements than the CGLS SM for the classical metrics (Bazzi et al., 2019).

Taking into account the importance of having accurate HR large scale SM datasets, in this study the $S^2MP$ algorithm was extended to provide SM estimates over both croplands and herbaceous vegetation at 1-km resolution, which also allowed to use Sentinel-3 (S3) NDVI instead of S2.

Two sets of SM maps at 1-km resolution were produced with the $S^2MP$ algorithm over six domains of $\sim 10^4$ km$^2$ in the southwest and southeast of France, Tunisia, North America, Spain and Australia (see panel A from Figure 1). A set is based on the combination of S1 and S2 measurements (S1+S2 maps), while the other is based on the combination of S1 and S3 measurements (S1+S3 maps). The $S^2MP$ S1+S2 and S1+S3 maps were compared to those provided by the SMAP+S1 product as well as those from the CGLS SM and SWI datasets from January to December 2019. The comparison was carried out on a per pixel basis and the results were analysed according to pixel homogeneity for areas covered by croplands and herbaceous vegetation. In addition, the HR time series were evaluated against in-situ measurements along with those of coarser resolution SM datasets from SMAP, SMOS, and ESA CCI.

The paper is structured as follows. Section 2 presents the different remotely sensed and ground based data that are used in this study. Section 3 describes the methodology used to estimate SM over croplands and herbaceous regions using S1+S3. Section 4 shows the S1+S3 $S^2MP$ maps and time series. They are also compared to the S1+S2 maps as well as to other HR datasets, coarse resolution datasets and in-situ measurements. Section 5 discusses the interest of the $S^2MP$ algorithm modifications and the remaining challenges for large scale HR SM mapping. Section 6 draws the conclusions of the study.

## 2  Data

### 2.1  Soil moisture maps computation

#### 2.1.1  Sentinel-1

The Sentinel-1 mission is the first satellites constellation mission of the Copernicus program and was conducted by ESA. The mission is composed of a constellation of two satellites sharing the same orbital plane. S1A was launched on 3 April 2014, and S1B on 25 April 2016. They were placed in a near-polar, sun-synchronous orbit. The revisit frequency is 12 days over Europe (6 days using both satellites) with crossing time at equator at 6:00 pm for the descending overpass. S1A and S1B carry onboard a C-band (wavelength  6 cm) SAR imaging instrument, enable to acquire imagery regardless of the weather and the time of the day.

For the production of the $S^2MP$ SM maps, S1A and S1B SAR images were collected over each region of study. S1 images are accessible from the Copernicus Open Access Hub. The S1 images (10 m x 10 m) were acquired in the Interferometric Wide-swath (IW) imagining mode with VV and VH polarizations and the S1 Toolbox (S1TBX) developed by ESA was used to calibrate the images. This calibration aims to convert digital number values from S1 images into backscattering coefficients ($\sigma°$) in a linear unit and ortho-rectifying the images using the Shuttle Radar Topography Mission (SRTM) Digital Elevation Model (DEM) at 30-m spatial resolution. A database of S1 images available from January to December 2019 was created for each region of study. The databases contain S1 images acquired both in ascending (afternoon) and descending (morning) modes.

#### 2.1.2  Sentinel-2

The Sentinel-2 A and B (S2A and S2B) satellites were launched on 23 June 2015 and 7 March 2017, and were placed in a near-polar, sun-synchronous orbit. The revisit frequency is 10 days (5 days with 2 satellites) and the descending orbit crossing time at equator is at 10:30 am. The spatial coverage ranges from 56° S to 84° N. The satellites carry onboard a multi-spectral instrument with 13 bands: 4 bands at 10-m, 6 bands at 20-m and 3 bands at 60-m spatial resolution. The orbital swath width is 290 km (Spoto et al., 2012).

For the production of the $S^2MP$ SM maps based on S1 and S2, optical images from S2A on dates close to S1 SAR images (less than 2 weeks) were downloaded from the French land data service center (Theia) website (https://www.theia-land.fr/). The S2A optical images (10 m x 10 m) are corrected for atmospheric effects and ortho-rectified.

#### 2.1.3  Sentinel-3

The Sentinel-3 A and B (S3A and S3B) satellites were launched on 16 February 2016 and on 25 April 2018, respectively. The S3 satellites orbit is a near-polar, sun-synchronous orbit with crossing time at equator at 10:00 am for the descending overpass. They carry onboard an optical instrument payload, the Ocean and Land Colour Instrument (OLCI), that samples 21 spectral bands ([0.4-1.02] $\mu$m) with a swath width of 1 270 km and a spatial resolution of 300 m. They also carry a dual-view scanning

temperature radiometer at 500-m spatial resolution: the Sea and Land Surface Temperature Radiometer (SLSTR). The revisit frequency of these instruments is 2 days when both satellites are used together (Donlon et al., 2012).

In this study, the S3 10-days synthesis NDVI at 1-km spatial resolution were used for the production of the $S^2MP$ SM maps based on S1 and S3. These data are accessible in the $SY\_2\_V10$ product (Henocq et al., 2018) and were downloaded from the Copernicus Open Access Hub. The data from this product rely upon the synergistic use of the OLCI and SLSTR

instruments. The product provides a 1-km VEGETATION-like product including 10-day synthesis surface reflectances and NDVI. The NDVI values correspond to a maximum NDVI value composite of all segments received for 10 days.

## 2.2 Data used for evaluation

### 2.2.1 Copernicus Global Land service

Two CGLS datasets were used to compare with the $S^2MP$ maps:

(*i*) The CGLS V111 S1 Surface Soil Moisture product (hereafter $CoperSSM$) is retrieved from the S1 radar backscatter images over the European continent at 1-km resolution (Bauer-Marschallinger et al., 2019). The images are acquired at C-band SAR in VV polarization, and the retrieval approach is based on a change detection algorithm (Bauer-Marschallinger et al., 2018). Changes observed in the C-band SAR backscatter coefficient are interpreted as changes in the SM values, whereas other surface properties such as the geometry, surface roughness and vegetation cover are assumed to be static in time for each pixel.

The algorithm provides local relative SM values in percentages ranging between 0% and 100% except in the case of extremely dry conditions, frozen soil, snow-covered soil and flooding. The data are sampled at 1-km resolution from 11°W to 50°E and from 35°N to 72°N.

(*ii*) The CGLS V101 S1 Soil Water Index product (hereafter $CoperSWI$) is derived from a fusion of surface SM observations from S1 C-band SAR and Metop ASCAT sensors (Bauer-Marschallinger et al., 2018). It uses a two-layer water balance

model that is adapted to use a recursive formulation and does not account for soil texture. A Surface State Flag (SSF) that indicates frozen/unfrozen/melting state of the surface, depending on the temperature, is used to identify SM values under non-frozen conditions to be used for the SWI calculation. SWI and quality flag values are calculated based on a phenomenological formulation that depends on the characteristic time length parameter (hereafter $T$). A large $T$ value represents an increase in reservoir depth or a decreased pseudo-diffusivity coefficient. This means that, for a fixed pseudo-diffusivity constant, an

increased $T$ value represents a deeper soil layer (Paulik et al., 2014). SWI estimations for eight different $T$ values are provided within the product. Previous evaluations of SWI data by Paulik et al. (2014) and Albergel et al. (2008) showed that the best agreement with in-situ measured surface SM is usually obtained with $T$ values in the range of 5-10, therefore SWI data with $T = 5$ were used in this study.

### 2.2.2 SMAP products

SMAP provides passive measurements of 1.4 GHz brightness temperatures in vertical and horizontal polarizations at a fixed incidence angle of 40 degrees with a resolution of $\sim 45$ km (Entekhabi et al., 2014) SMAP ascending and descending orbits

cross the equator at 6:00 pm and 6:00 am respectively, and the maximum revisit period is 3 days. Several HR and CR SM datasets from SMAP were used for the evaluation of the $S^2MP$ maps:

(*i*) The SMAP L3 V6 SM product (hereafter $SMAPL3$). It is a daily gridded composite of the SMAP L2 V5 SM files (O'Neill et al., 2018, 2019b). Only SM estimates derived from L1C brightness temperatures (Chan et al., 2018) using the Single Channel Algorithms V-polarization (Entekhabi et al., 2010) were considered. SMAP L3 data are sampled at 36-km resolution.

(*ii*) The SMAP Enhanced L3 V1 SM product (hereafter $SMAPL3E$), which is obtained by oversampling the L1C brightness temperatures from 36-km to 9-km resolution using an interpolation algorithm (O'Neill et al., 2019a). Only SM estimates derived using the Single Channel Algorithms V-polarization were considered.

(*iii*) The SMAP+S1 L2 V1 SM product (hereafter $SMAPS1$) provides SM at 1-km resolution that are estimated using the SMAP Enhanced L3 V004 Half-Orbit at 9-km resolution and Copernicus S1A and S1B C-band SAR data (Das et al., 2020). Brightness temperatures from SMAP are disaggregated on the 1-km EASE-Grid by using the S1 radar backscatter data and HR SM estimates are obtained using the SMAP Active-Passive algorithm. The closest data in time between descending and ascending orbits from SMAP are used to spatially match up with the S1 scene.

### 2.2.3 SMOS

The SMOS mission (Kerr et al., 2010) carries a passive interferometric radiometer operating at L-band (21 cm, 1.4 GHz) with a spatial resolution of 25-50 depending on the position on the field of view (43 km on average). The following CR SM datasets from SMOS were used in this study:

(*i*) The CATDS SMOS L3 V7 SM product (hereafter $SMOSL3$), which is a multi-orbit SM product provided by the Centre Aval de Traitement des Données (CATDS) with a grid resolution of 25 km (Al Bitar et al., 2017). The SM retrieval process is based on the algorithm used for the SMOS L2 product (Kerr et al., 2012) but using simultaneously three orbits within a one week period to better constrain the SM and optical depth estimations.

(*ii*) The ESA SMOS Near Neal Time (NRT) Neural Network (NN) V2 SM product (hereafter $SMOSNRT$), provided on the icosahedral equal area grid (ISEA4H9) with 15-km resolution (Rodríguez-Fernández et al., 2017). It is designed to provide SM in less than 3.5 h after sensing. The algorithm uses a NN trained using SMOS L2 SM data (Kerr et al., 2012). The input data for the NN are SMOS brightness temperatures with incidence angles from 30 to 45 degrees for horizontal and vertical polarizations and soil temperature in the 0–7 cm layer from the European Centre for Medium-Range Weather Forecasts (ECMWF) models.

### 2.2.4 ESA CCI COMBINED product

In the COMBINED product of ESA SM CCI V5.2 (hereafter $CCISM$, Dorigo et al., 2017; Gruber et al., 2019) L2 datasets from different active and passive sensors are directly scaled by matching their Cumulative Density Functions (CDF) to that of the Global Land Data Assimilation System (GLDAS, Rodell et al., 2004) Noah land surface model in order to remove relative biases and harmonize their dynamical ranges. In the period of this study, ESA CCI Combined uses the H SAF active sensor

products from the Advanced Scatterometer A and B (ASCAT, Wagner et al., 2013) and the passive sensor products from the Advanced Microwave Scanning Radiometer 2 (AMSR2, Kim et al., 2015; Imaoka et al., 2000), as well as those from SMAP and SMOS. SM data from the passive sensors are estimated using the Land Parameter Retrieval Model (LPRM) V6 (Van der Schalie et al., 2016, 2017). The data are sampled at 25-km resolution.

### 2.2.5 Land cover

The Copernicus Global Land Service (CGLS) V3 Dynamic Land Cover Map product delivers a global land cover map at 100-m resolution covering the period between 2015 to 2019 (Buchhorn et al., 2020). For each year, a land cover map is provided with three different levels of classes: 11 classes at level 1 (all types of forests are considered as an unique land cover class), 13 classes at level 2 (forests are splitted in two land cover classes: open and closed forests) and up to 22 classes at level 3 (all types of open and closed forests are considered). In this study, only the 2019 land cover map at level 1 was considered. The panel B of Figure 1 shows the 7 land cover classes that are represented in the 6 regions of study. In this figure, the land cover map was aggregated from 100-m to 1-km resolution for evaluation purposes meaning that only the dominant land cover for each 1-km$^2$ pixel is shown.

### 2.2.6 In-situ measurements

The evaluation against in-situ measurements of soil moisture was performed using data from the REMEDHUS (Gonzalez-Zamora et al., 2018), SMOSMANIA (Calvet et al., 2007), OZNET (Smith et al., 2012; Young et al., 2008), USCRN (Bell et al., 2013), ARM (Cook, 2016, 2018) and the Merguellil networks (Amri et al., 2011; Gorrab et al., 2015) that are located within the six regions of this study (Table 1). All the data, except those from the Merguellil network, were retrieved from the International Soil Moisture Network (ISMN, Dorigo et al., 2011, 2021). Only sensors between 0 and 5 cm depth were considered. In total, 65 ISMN and 5 Merguellil sites were used for the scaling of the $CoperSSM$ and $CoperSWI$ data (Sect. 3.2). Less sites (40 from ISMN and 2 from Merguellil) were used to assess the remotely sensed data following the criteria explained in Section 3.3. The different in-situ stations can be located using the panels A and B from Figure 1.

## 3 Methods

### 3.1 Building $S^2MP$ maps using NDVI derived from Sentinel-3

The $S^2MP$ algorithm (El Hajj et al., 2017) was originally designed to estimate surface SM at the scale of agricultural plots (10-m resolution) using synergistically data derived from S1 radar signal and S2 optical images as inputs to a neural network. The neural network was first trained using a synthetic database gathering *(i)* SAR C-band backscatter coefficients in the VV polarization *(ii)* incidence angles (from 20 to 45 degrees), and *(iii)* NDVI as inputs and SM examples as target. This synthetic database was built using a Water Cloud Model (Baghdadi et al., 2017) combined with an Integral Equation Model (Baghdadi et al., 2006, 2011) that was specially modified and optimized for this application. Then, the $S^2MP$ algorithm was applied to a

real database gathering the SAR backscatter coefficient in VV polarization from S1, the incidence angle of the SAR acquisitions
and the NDVI derived from optical images taken by S2 as follows. Firstly, the NDVI was computed at 10-m spatial resolution
(native resolution of S2) using the atmospherically and ortho-rectified S2 images. To overcome the cloud cover issue present
in optical images, a gap filling was performed using the linear interpolation to obtain two cloud-free NDVI images per month
(1st and 15th of each month). To derive S2 NDVI data at the S1 acquisition dates, a linear interpolation was performed for each
S2 pixel using the NDVI values corresponding to the closest S2 images acquired before and after the S1 date. Secondly, the
10-m resolution S1 backscattering signal, incidence angle and S2 NDVI were averaged for each 100-m pixels from the CGLS
land cover map. Then, the SM estimation using the $S^2MP$ algorithm was performed at 100-m spatial resolution over pixels
covered by croplands using the CGLS land cover map described above (see Sect. 2.2.1). There is no retrieval for other types of
land cover.

In contrast to the (El Hajj et al., 2017) approach described above, in the current study S1+S2 maps were also computed
for 100-m pixels covered by herbaceous vegetation. In addition, the 100-m SM estimations were aggregated to 1-km. On the
other hand, HR SM maps were also produced using NDVI from the $SY\_2\_V10$ product at 1-km resolution from S3 (see Sect.
2.1.3). Due to the spatial resolution of the S3 NDVI, the S1 backscattering signal and incidence angle were first aggregated
from 10-m to 100-m resolution and then re-aggregated from 100-m to 1-km only over croplands and herbaceous vegetation.
Then, the neural network was only applied over 1-km$^2$ pixels that are partly or entirely covered by croplands and/or herbaceous
vegetation. The objective was to assess the impact of using lower spatial resolution NDVI as inputs to the $S^2MP$ algorithm.

Hereafter $S^2MP_{S1S2}$ and $S^2MP_{S1S3}$ will refer to the $S^2MP$ SM datasets derived based of the synergistic use of S1 + S2,
and S1 + S3 measurements, respectively. $S^2MP_{S1S2}$ and $S^2MP_{S1S3}$ were produced from January to December 2019.

Both for $S^2MP_{S1S2}$ and $S^2MP_{S1S3}$, it is important to highlight again that there is no SM estimate available over 1-km$^2$
pixels not covered at all by croplands and herbaceous vegetation. However, as long as there is a fraction of croplands and/or
herbaceous vegetation (whatever the amount) within the pixel, SM values are provided. The proportion of croplands and
herbaceous vegetation within each 1-km$^2$ pixel for the 6 regions of study is shown on the panel C from Figure 1.

## 3.2  CoperSSM and CoperSWI rescaling

Relative SM indices from $CoperSSM$ and $CoperSWI$ were scaled against in-situ measurements for each region indepen-
dently. This process is needed to transform the indices into SM estimates with volumetric units (m$^3$ m$^{-3}$). The following
scaling formula was applied:

$$SM_n^* = SM_n \times [max(SM_n^{IS}) - min(SM_n^{IS})] + min(SM_n^{IS}) \tag{1}$$

where $SM_n$ and $SM_n^*$ are respectively the original and scaled SM indices from $CoperSSM$. $SM_n^{IS}$ includes all the SM
measurements from all the in-situ time series available for the current region $n$ in 2019 (Table 1). This concretely means 19
in-situ time series for the Spanish region, 4 in the southwest of France, 5 in the southeast of France, 11 for the Australian
region, 26 in North America, and 5 in Tunisia.

The 2.5% lowest and 2.5% highest values are discarded before applying the minimum and maximum functions to remove the effect of possible outliers that can be caused by instrumental noise (Brocca et al., 2011). The same process was also undertaken to scale the SWI values from $CoperSWI$. The Copernicus indices were also scaled using $SMOSL3$ or $SMAPL3$ to obtain the maximum and minimum references instead of in-situ measurements. The final results were quite comparable regardless the reference used and thus only the scaling against in-situ measurements was used for the rest of the study.

### 3.3 Datasets comparisons

Comparisons between datasets and evaluations against in-situ measurements were done from January to December 2019. In a first step, the $S^2MP_{S1S2}$ and $S^2MP_{S1S3}$ maps were compared on a per pixel basis for each region in terms of Pearson correlation ($R$), bias and standard deviation of the difference ($STDD$, also referred to as unbiased Root Mean Square of the Difference by some authors). The metrics for which the P-value exceeded the threshold of 5% (interval of confidence of 95%) were discarded.

In a second step, $S^2MP_{S1S3}$ were compared to the 3 HR datasets described in Section 2: $CoperSSM$, $CoperSWI$ and $SMAPS1$. This analysis was also performed by computing $R$, bias and $STDD$ on a per pixel basis for each region (all the HR datasets are sampled on the same 1-km regular grid). In addition, the metrics were analysed as a function of croplands and herbaceous vegetation coverage over 1-km$^2$ pixels. The metrics for which the P-value exceeded the threshold of 5% were discarded.

In a third step, all the different CR and HR datasets were evaluated against the in-situ measurements available in the 6 regions of study. To perform the analysis in the optimal conditions, only morning orbits from SMOS (ascending overpasses) and SMAP (descending overpasses) within this time period were considered. During the night and early in the morning, the soil is in thermal balance, meaning that the vegetation temperature is equal to the soil temperature. During the afternoon, the balance is lost and the vegetation temperature is closer to the air temperature leading to satellite estimates of lower quality. This is often reflected by lower performances against in-situ measurements for the afternoon SM estimates than those of the morning (Leroux et al., 2014). For each ground station, the closest time series from each remotely sensed dataset was compared to the in-situ measurements by computing $R$, bias and $STDD$. Only samples for which the difference in acquisition times with the in-situ measurements does not exceed 1 hour were taken into account to compute those statistical metrics. Metrics for which the corresponding P-value exceeded the threshold value of 5% were discarded. This implies that only in-situ locations where all the comparisons between the remotely sensed and in-situ time series showing significant metrics were considered for the assessment (Table 1). This concretely means 13 in-situ time series for the Spanish region, 3 in the southwest of France, 0 in the southeast of France, 10 for the Australian region, 15 in North America, and 2 in Tunisia.

Then, remotely sensed time series of anomalies in a 35-days time window were also compared to those of the in-situ measurements in terms of $R$. They were derived as follows.

$$SM_t^a = (SM_t - \mu_t)/\sigma_t$$

$$\mu_t = 1/N \times \sum_{n=t_1}^{t_2} SM_n \qquad\qquad (2)$$

$$\sigma_t = \sqrt{1/(N-1) \times \sum_{n=t_1}^{t_2} (SM_n - \mu)^2}$$

where $SM_t^a$ and $SM_t$ are the SM and anomalies values at time $t$, respectively. N is the number of observations from $t$ minus 17 days ($t_1$) to $t$ plus 17 days ($t_2$).

Finally, the HR datasets ($S^2MP_{S1S2}$, $S^2MP_{S1S3}$, $CoperSSM$, $CoperSWI$ and $SMAPS1$) were evaluated against in-situ measurements ($R$, bias and $STDD$ with P-values below 5%) after aggregation at 25-km resolution (same grid as that of $CCISM$) in order to compare their performances to the CR data at a comparable resolution.

## 4 Results

### 4.1 Sentinel-3 versus Sentinel-2 NDVI

S2 and S3 NDVI were compared in each region of study at 1-km grid scale in terms of $R^2$ as scatter plots in Figure 2. High $R^2$ are observed in Australia (0.86) and Tunisia (0.79), and moderate values are found in North America (0.68) and Spain (0.64). No significant correlation is observed in the southwest and southeast of France The results indicate that in dry regions such as in Australia, Tunisia and North America, high correlation exists between S2 and S3 NDVI whereas low correlation is present in temperate areas with patchy land covers such as in the south of France. For all the study regions, S3 NDVI saturates between 0.6 and 0.7 whereas S2 NDVI reaches higher values between 0.8 and 0.9. The difference could be mainly due to the mixture of surface reflectances from different land cover classes within the 1-km S3 NDVI.

Figure 3 shows the distribution of $R^2$ between S2 and S3 NDVI as a function of the months for each region. In general, higher correlations are obtained in the summer season (dry periods) than in winter and spring (humid periods). For example, $R^2$ between S2 and S3 NDVI is only high (0.72) from January to June (summer and autumn seasons) in Australia. In North America, from March to July, $R^2$ is between 0.25 and 0.53 whereas no significant correlation between S2 and S3 NDVI is found for the others months. In the southwest and southeast of France, no correlation are found for most of the months except in summer such as in June, July and August. On the one hand, the highest NDVI values are found in winter and spring seasons due to the development of the vegetation cycles. On the other hand, summer seasons usually show lower NDVI values that corresponds to bare soil conditions except in the presence of irrigated summer crops. Thus, S2 and S3 NDVI are highly correlated for low NDVI values (usually in summer). However, the correlation decreases for high NDVI values because of the peak of the vegetation development (in spring).

### 4.2 $S^2MP_{S1S3}$ comparison to $S^2MP_{S1S2}$

Figure 4 shows $R$, bias and $STDD$ between $S^2MP_{S1S2}$ and $S^2MP_{S1S3}$ for the 6 study regions. A very good agreement between the two datasets was found in all the regions with $R \geq 0.9$, bias $\leq 0.04$ m$^3$ m$^{-3}$ ($S^2MP_{S1S3}$ minus $S^2MP_{S1S2}$) and $STDD \leq 0.03$ m$^3$ m$^{-3}$ for most of the areas. However, some differences in terms of bias can be seen between the two datasets in the northwest of the Spanish region (Fig 4c), in the areas with significant forests cover in the France southwest region (Fig 4i) and in narrow areas of the Tunisian region (Fig 4f).

The somewhat higher differences in the Spanish and France southwest regions are seen over pixels covered by forests with a small fraction of croplands and herbaceous vegetation (see the Spanish region in panels B and C from Fig. 1). The somewhat larger differences in some narrow areas of Tunisia is due to heterogeneous land cover around several river basins with rolling topography, sparse forests as well as grasslands.

As discussed above, these small differences were expected due to the differences seen between S3 and S2 NDVI (Sect. 4.1) and the different way of aggregating the S1 backscatter coefficients (Sect. 3.1). However, taking into account the overall very good agreement between $S^2MP_{S1S2}$ and $S^2MP_{S1S3}$, for the sake of simplicity and clarity, in Section 4.3 only $S^2MP_{S1S3}$ is compared to the other HR datasets.

### 4.3 General comparison of $S^2MP_{S1S3}$ against the HR SM datasets

Figures 5, 6 and 7 present the comparison of $S^2MP_{S1S3}$ against $CoperSSM$, $CoperSWI$ as well as $SMAPS1$ over the 6 study regions in terms of bias, $STDD$ and $R$, respectively. Some diagonal structures can be seen in the maps comparing $S^2MP_{S1S3}$ to $CoperSSM$ in Spain and in the southwest of France. These artifacts, most pronounced in the correlation maps but also present in the bias and $STDD$ maps, come from the $CoperSSM$ data as previously discussed by Bazzi et al. (2019). Indeed, the artifacts are seen on the sub-swaths of the S1 product showing a big difference between the SM estimations in the $CoperSSM$ at the same SM estimation date. Bazzi et al. (2019) showed that the difference of the SM estimation at both sides of the sub-swath at a given date of the $CoperSSM$ map can reach 0.11 m$^3$m$^{-3}$.

#### 4.3.1 Comparison of the order of magnitude

According to Figure 5, $S^2MP_{S1S3}$ shows a bias in the range from -0.1 to 0.1 m$^3$ m$^{-3}$ with respect to the other HR products over most of the pixels within the 6 regions of study. However, there are areas in the Spanish, Tunisian and France southeast domains, where $S^2MP_{S1S3}$ shows a dry bias of absolute value larger than 0.1 m$^3$ m$^{-3}$. This is also particularly the case in the southwest of France, with respect to $CoperSSM$ and $CoperSWI$ as well as with respect to $SMAPS1$ in Australia and North America. For these regions and HR datasets, the bias is negative over the whole area. For all the other combinations of regions and HR products, the bias values are both positive and negative. In general, there is no clear relationship between the sign of the bias and the dominant land cover class. However, in the case of the comparison between $S^2MP_{S1S3}$ and $SMAPS1$ in the France southwest region, the bias distribution is splitted in two (Fig. 5i). A wet bias is observed in the west part of the region corresponding to forests areas with low fractions of croplands and herbaceous vegetation, while a dry bias is found in the east

part corresponding to areas dominated by croplands (Panels B and C from Fig. 1). In addition, the dry bias observed in the east part of the Tunisian region corresponds to an area of salted lakes, named Sebkha, whose water and moisture contents can vary significantly according to climate.

Figure 6 shows that the $STDD$ values of $S^2MP_{S1S3}$ with respect to $CoperSSM$ and $CoperSWI$ are lower than 0.06 m$^3$ m$^{-3}$ over almost all the pixels within the 4 regions where the Copernicus datasets are available. Higher values close to
340 0.08-0.10 m$^3$ m$^{-3}$ between $S^2MP_{S1S3}$ and $CoperSSM$ are found in the southwest part of the southwest region of France. The $STDD$ obtained between $S^2MP_{S1S3}$ and $SMAPS1$ are comparable to those obtained with respect to the Copernicus datasets. However, values reaching 0.08-0.12 m$^3$ m$^{-3}$ are more often found, in particular in the west part of the Spanish region, and sparsely in the southwest and southeast of France. In Australia and North America (Fig. 6m,n), the $STDD$ with respect to $SMAPS1$ are quite similar to those found in Spain, Tunisia and France (Fig. 6c,f,i,l), where values can reach $\sim 0.08 - 0.12$
m$^3$ m$^{-3}$ in the southeast and west parts of the Australia and North America regions, respectively. There is no clear and unique relationship with the dominant land cover class. For instance, the $STDD$ with respect to $SMAPS1$ in the southwest of France is higher over the forests than over the croplands dominated areas, while in the North America region the $STDD$ was found to be lower over the forests (see panels B and C from Fig. 1).

### 4.3.2 Comparison of the temporal dynamics

Overall, $S^2MP_{S1S3}$ and $CoperSSM$ show a high correlation (above 0.7-0.8) over almost all the pixels within all the regions of study (Figs. 7a,d,g,j). In contrast, lower values are found for the correlation between $S^2MP_{S1S3}$ and $CoperSWI$ (Figs. 7b,e,h,k) as well as between $S^2MP_{S1S3}$ and $SMAPS1$ (Figs. 7c,f,i,l,m,n). $R$ rarely exceed 0.7 and values lower than 0.6 are observed in many large areas.

In the Spanish region, the highest $R$ values are obtained in the areas dominated by croplands. The lowest values are found
in the northwest over heterogeneous pixels dominated by forests (Panels B and C from Fig. 1). Similar spatial features are observed in the 3 maps comparing $S^2MP_{S1S3}$ to $CoperSSM$, $CoperSWI$ and $SMAPS1$ (Figs. 7a,b,c). However, lower $R$ values are found with respect to $SMAPS1$ and $CoperSWI$. In addition, the comparison against $CoperSWI$ shows $R$ below 0.5 in a few spots in the south and the center of the region.

In Tunisia (Fig. 7d-f), the correlation values obtained in the north are quite good with values of 0.8-0.9 with respect to
360 $CoperSSM$. $R$ drop in the southeast and southwest to values lower than 0.5. The decrease in the southwest can be partly explained by the proximity of coasts, where mixed land cover pixels include urban areas (Panel B from Fig. 1). The correlation with respect to $CoperSWI$ (Fig. 7e) is only higher than 0.5 for the regions where the 1-km$^2$ pixels are dominated by croplands. The comparison between $S^2MP_{S1S3}$ and $SMAPS1$ results in a large range of correlations. Only the very north areas dominated by croplands show correlations above 0.6 (Panels B and C from Fig. 1). $R$ close to 0.4-0.5 are found in the east
and west parts of the region. Values lower than 0.2 are observed in the center of the region over heterogeneous pixels around several river basins that were also highlighted in Section 4.2 with Figure 4f.

In the France southwest region (Fig. 7g-i), the distribution of $R$ is quite homogeneous over the whole area and does not vary significantly according to pixels dominated by croplands or by forests (Panel B from Fig. 1). $R$ values with respect to

$CoperSSM$ are mainly above 0.8 over most of the pixels while the values drop to 0.5 with respect to $CoperSWI$. The comparison between $S^2MP_{S1S3}$ and $SMAPS1$ shows $R$ closer to 0.6 in general but really low correlation values (below 0.2) appear over several pixels. The same pattern is observed in the France southeast region (Fig. 7j-l), but correlations are only significant over areas dominated by croplands.

$R$ between $S^2MP_{S1S3}$ and $SMAPS1$ reach 0.7-0.8 in Australia and North America (Fig. 7m,n) with no clear relationship with the land cover type.

### 4.3.3 Comparison over areas dominated by croplands and herbaceous vegetation

To get further insight into the analysis over croplands and herbaceous vegetation, $S^2MP_{S1S3}$ was exclusively compared to $CoperSSM$, $CoperSWI$ and $SMAPS1$ over pixels where one of these two land cover classes is dominant. For each region and land cover, a set of metrics ($R$, bias and $STDD$) is computed in two ways. One set is computed by only taking into account pixels covered by less than 75% of croplands or herbaceous vegetation. The other set is computed by only taking into account pixels covered by at least 75% of croplands or herbaceous vegetation. The results are summed up in Table 2.

Over pixels in Europe (Spain, Tunisia, France) where croplands represent less than 75% of the area, $S^2MP_{S1S3}$ is better correlated to $CoperSSM$ than $CoperSWI$ and $SMAPS1$. In general, high $R$ values are found in Spain (0.54-0.63), moderate values (0.28-0.63) are observed in France and low values are found in Tunisia (0.37-0.38). In addition, $R$ values obtained in Australia and North America with respect to $SMAPS1$ are comparable to those found in Spain. Absolute bias between $S^2MP_{S1S3}$ and the 3 HR datasets in Spain ( 0.01 m$^3$ m$^{-3}$) are lower or similar to those found in the other regions. The strongest absolute bias is observed in North America with respect to $SMAPS1$ with 0.07 m$^3$ m$^{-3}$. According to the $STDD$ values, no particular trend is observed over the 6 regions of study and values range from 0.04 to 0.09 m$^3$ m$^{-3}$. For most of the regions and comparisons, the correlation values significantly increase (+ 0.05-0.1) over pixels that contain at least 75% of croplands. In overall, absolute bias and $STDD$ values remain similar but sometimes there is a slight decrease (-0.01 m$^3$ m$^{-3}$).

Taking into account pixels where herbaceous vegetation represent less than 75% of the area, $S^2MP_{S1S3}$ is only better correlated to $CoperSSM$ in Spain and the southwest of France. In general, high $R$ values are found in Spain (0.64-0.67), moderate values (0.36-0.69) are observed in France and low values are found in Tunisia (0.32-0.36). In addition, $R$ values obtained in Australia and North America with respect to $SMAPS1$ are comparable to those found in the southwest of France. Absolute bias between $S^2MP_{S1S3}$ and the 3 HR datasets in Tunisia (0.01-0.02 m$^3$ m$^{-3}$) are lower or similar to those found in the other regions. The strongest absolute biases are observed in the southwest of France with 0.09 m$^3$ m$^{-3}$ for $CoperSWI$ and $SMAPS1$, respectively. The $STDD$ is higher with respect to $SMAPS1$ and can reach 0.07, 0.08 and 0.10 m$^3$ m$^{-3}$ in Tunisia, Spain, and in the southwest of France, respectively. In contrast with croplands, the correlation values does not systematically increase over pixels that contain at least 75% of herbaceous vegetation. There is no significant change concerning the $STDD$ and in overall the absolute bias increases with respect to almost all the HR datasets (+0.01-0.04 $^3$ m$^{-3}$).

It is noteworthy that the bias is significantly higher over herbaceous vegetation than over croplands in Spain and in the southwest of France, while the $STDD$ is quite similar regardless the region of study. In addition, higher $R$ values are found over herbaceous vegetation in Spain. In Tunisia, the correlation and absolute bias values are among the lowest both over

croplands and herbaceous vegetation. Low absolute biases can be partly explained by the fact that the region is really dry with desert areas implying a small SM dynamic range with very low values regardless the estimation algorithm used.

## 4.4  Evaluation against in-situ measurements

Table 4 presents the evaluation of the different CR and HR SM products with respect to in-situ measurements in terms of bias, $STDD$ and $R$ of the original time series as well as Pearson correlation of the anomalies time series ($R^a$). In addition, Figure 8 shows examples of time series of the different HR and CR datasets at 6 in-situ stations used in this study (one for each region).

The highest absolute biases with respect to in-situ measurements are obtained for $S^2MP_{S1S2}$ and $S^2MP_{S1S3}$ with -0.06 m$^3$ m$^{-3}$, closely followed by $CoperSWI$ with 0.05 m$^3$ m$^{-3}$. $SMOSNRT$, $SMOSL3$, $CCISM$ and $SMAPS1$ show the lowest absolute biases with 0.03 m$^3$ m$^{-3}$.

The highest $STDD$ with respect to in-situ measurements are obtained for $CoperSSM$ (0.08 m$^3$ m$^{-3}$) and $SMOSL3$ (0.07 m$^3$ m$^{-3}$). The other datasets shows comparable $STDD$ with 0.05 or 0.06 m$^3$ m$^{-3}$.

In general, higher correlation values are obtained for the CR data (0.67-0.77) than for the HR data (0.53-0.74). The lowest correlations are found for the Sentinel-only HR datasets with 0.53 for $CoperSSM$, 0.56 for $S^2MP_{S1S3}$ and 0.59 for $S^2MP_{S1S2}$. Concerning the HR data obtained from merging approaches, $SMAPS1$ still shows a value lower than the CR datasets with 0.64 but $CoperSWI$ shows the third best value with 0.74, just after $SMAPL3E$ (0.77) and $SMAPL3$ (0.76).

Regarding the correlation of the anomalies time series, $SMAPL3E$ and $SMAPL3$ get the highest $R^a$ with respect to in-situ measurements with 0.59 and 0.58. $CoperSSM$, $S^2MP_{S1S2}$, $SMAPS1$, and $S^2MP_{S1S3}$ show the lowest $R^a$ with 0.18, 0.36, 0.35 and 0.37 respectively.

The CR time series have a temporal revisit roughly five times higher than those from $S^2MP_{S1S2}$, $S^2MP_{S1S3}$, $CoperSSM$ and $SMAPS1$ (Fig. 8). In order to understand if the low temporal revisit of the HR data affects their performances against in-situ measurements, one observation out of five was removed from the CR time series and the metrics were re-computed (not presented). However, no significant differences in terms of $R$, bias and $STDD$ were found.

The performances of the two Sentinel-only HR datasets averaged at 25-km resolution ($S^2MP^*_{S1S2}$, $S^2MP^*_{S1S3}$, $CoperSSM^*$) with respect to in-situ measurements are comparable to the performances obtained for the original 1-km datasets ($S^2MP_{S1S2}$, $S^2MP_{S1S3}$, $CoperSSM$,$CoperSWI^*$). In contrast, for the $SMAPS1^*$ dataset, which is a downscaled product, the correlation increases from 0.64 at 1-km resolution ($SMAPS1$) to 0.79 at 25-km resolution, which is the highest correlation found among all the datasets. In addition, the $STDD$ and bias slightly decrease. $R^a$ also increases from 0.35 at 1-km resolution to 0.44 at 25-km resolution, but it does not reach the values of $R^a$ obtained for the SMAP-only products ($SMAPL3$ and $SMAPL3E$ with 0.58-0.59).

## 5  Discussion

$S^2MP$, $S^2MP_{S1S3}$, $CoperSSM$, $CoperSWI$ and $SMAPS1$ are all HR datasets that were produced with different approaches. Two products were obtained by merging S1 data with ASCAT ($CoperSWI$) and SMAP ($SMAPS1$), respec-

435 tively. $S^2MP$ and $CoperSSM$ are based on Sentinel-only. The last one is computed from local temporal variations of the S1 backscatter coefficients time series following the method of Wagner (1998). In contrast, $S^2MP$ provides SM estimates derived from a NN that uses a database gathering backscatter coefficients and HR NDVI from Sentinel as inputs. The NN was initially trained on a synthetic database containing backscatter coefficients and surface characteristics such as SM and vegetation status (approximated by NDVI) that were predicted from electromagnetic modelling.

Initially, the $S^2MP$ algorithm by El Hajj et al. (2017) was only providing SM estimates over croplands at 10-m resolution using NDVI derived from S2 optical images. In the framework of this study, the algorithm has been extended to provide SM estimates at 1-km resolution also over herbaceous vegetation areas. and S2 was replaced by S3. However, despite the different ways of aggregating the S1 radar signal and the differences between S3 and S2 NDVI (Sect. 4.1 with Figs. 3 and 2), $S^2MP_{S1S2}$ and $S^2MP_{S1S3}$ are in very good agreement over the 6 study regions (Sect. 4.2 with Fig. 4). Thus, these results imply that it is

possible to replace S2 by S3 in the $S^2MP$ approach without losing skill. In addition, although the higher temporal revisit of S3 compared to S2 does not allow the $S^2MP$ algorithm to provide SM estimates more frequently because the effective revisit is that of S1 (see Fig. 8), the production of the SM daily maps using S3 instead of S2 is easier and faster. S2 and S3 NDVI are derived from optical measurements remotely sensed from space that are highly dependent on the cloud cover situation. Both S2 or S3 NDVI can be unreliable during long rainy or cloudy periods ($> 15$ days) over specific regions. However, the higher

temporal revisit of S3 allows the instruments onboard S3 to retrieve more optical images without cloud conditions than those onboard S2. This results in a better estimation of the vegetation cycle through the NDVI computation. Finally, less processing steps are required thanks to the availability of NDVI estimates already provided in the 1-km VEGETATION-like product from Copernicus (see Sec. 2.1.3).

According to Bazzi et al. (2019), the $S^2MP$ algorithm tends to provide unreliable SM estimates when the NDVI used

exceeds 0.7. NDVI above this value correspond to well-developed vegetation and even if it is more common to have NDVI lower than 0.7 with S3 than with S2 (Figure 2) due to averaging effets, using S3 NDVI does not solve the problem. Indeed, in the particular cases of well-developed vegetation, the problem does not arise from the S2 of S3 NDVI itself, but from the C-band SAR signal which fails to penetrate the vegetation cover.

Regarding the comparisons between the HR datasets carried in Section 4.3, $S^2MP_{S1S3}$ shows a temporal dynamic closer

to those of $CoperSSM$, $CoperSWI$ and $SMAPS1$ over semi-dry areas such as in Spain, North America and Australia. However, the correlation drops significantly over very dry zones (Tunisia). Over semi-humid areas (France), the temporal dynamic between $S^2MP_{S1S3}$ and $CoperSSM$ is in better agreement than with respect to $CoperSWI$ and $SMAPS1$. The order of magnitude (bias and $STDD$) between $S^2MP_{S1S3}$ and the other HR datasets is quite similar regardless the study region, the land cover as well as the climate zone (rather dry or humid). In addition, it is noteworthy to highlight that differences

in terms of bias and $STDD$ are not systematically reduced over homogeneous pixels covered by croplands or herbaceous vegetation. It means that inherent biases exist between the algorithms and they might persist over other land cover classes such as over forests (if the $S^2MP$ approach is extended to forests).

At the moment, by construction, the $S^2MP$ algorithm starts being out of its application domain when considering pixels dominated by forests cover. This is also the case for the change detection approach use to produce $CoperSSM$. Indeed, the

SM indices computation does currently not account for vegetation dynamics, which can lead to biases over areas covered by seasonal and dense vegetation. In addition, for most applications the $CoperSSM$ indices as well as those from $CoperSWI$ should be transformed into SM time series and this will be problematic without reference SM values under forest to scale them. Therefore, an extension of the $S^2MP$ algorithm to forest areas would definitely be interesting to provide HR SM mapping over large regions inside and outside Europe.

In this study, most of the SM measurements used from the ground stations were representative of croplands and herbaceous regions. Hence, the relative performances of the HR datasets ($S^2MP$, $CoperSSM$, $CoperSWI$, $SMAPS1$) were not assessed over dense vegetation areas.

In addition, slightly better results of $S^2MP_{S1S3}$ with respect to in-situ measurements compared to those of $CoperSSM$ were found, except for the bias (Table 4).

SM estimates using the $S^2MP$ algorithm were already evaluated against in-situ measurements along with other HR and CR datasets by El Hajj et al. (2018) and by Bazzi et al. (2019). In El Hajj et al. (2018), the authors found that $S^2MP_{S1S2}$ shows lower correlation with respect to in-situ measurements than $SMAPL3$ and $SMAPL3E$ but higher than $SMOSNRT$, $SMOSL3$ and $SMAPS1$. In contrast, in the current study, the $S^2MP_{S1S2}$ shows a lower correlation against in-situ measurements than all the other HR and CR products. However, in El Hajj et al. (2018), SM estimates from the $S^2MP$ algorithm were only derived over croplands while in our study, the SM estimation was performed both over croplands and herbaceous vegetation. Moreover, their analysis was only carried out in the south of France during a different time period (from January 2016 to June 2017). In Bazzi et al. (2019), the authors found that $S^2MP_{S1S2}$ is better correlated to in-situ measurements than $CoperSSM$. According to the results of our study (Table 4), higher correlations are also obtained for $S^2MP_{S1S2}$. In addition, it is interesting to note that $S^2MP_{S1S2}$ and $S^2MP_{S1S3}$ show similar performances with respect to in-situ measurements.

Taking into account the evaluations of all the HR and CR products together (Table 4), HR merged datasets ($SMAPS1$, $CoperSWI$) provide better estimations or temporal agreement with respect to in-situ measurements than the HR Sentinel-only ones ($S^2MP_{S1S2}$, $S^2MP_{S1S3}$, $CoperSSM$). However, they still show similar or lower performances than the CR datasets (Tab. 4). This can be partly explained by the fact that the HR datasets provide SM estimates using C-band measurements while the CR datasets used in this study are computed using L-band measurements. Indeed, SMOS and SMAP were specifically designed to measure surface SM, which was not the case for the Sentinel satellites. Regarding the higher performances of the CR datasets over the HR ones with respect to in-situ measurements, Bauer-Marschallinger et al. (2019) also demonstrated that 25-km resolution SM estimates from ASCAT were better correlated to in-situ measurements within the Italian Umbria region that those of the 1-km resolution $CoperSSM$. The results of our study are also in perfect agreement with the findings by Ojha et al. (2021), who showed over several regions in France and Spain that two merged products, $SMAPS1$ and SMAP + DISPATCH (Merlin et al., 2012), were better correlated to in-situ measurements than $CoperSSM$. Finally, it is noteworthy that the performances of $SMAPS1$ aggregated from 1-km to 25-km resolution (CR) increase to values similar to those of $SMAPL3$. This implies that the gain in resolution brought by merging data of different resolutions comes at the expense of introducing uncertainties in the resulting HR dataset.

Obviously, this study was limited to comparisons over 6 regions of $10^4\ km^2$ within a 1-year time period, so the results can not be straightforwardly extended to a global scale. However, the results of the study shows that the use of S3 NDVI as input to the $S^2MP$ algorithm leads to SM estimates comparable to those obtained with S2 NDVI. Howevere, HR SM estimates does not necessary lead to better performances with respect to in-situ measurements than CR SM estimates. In particular, retrieval algorithms only based on Sentinel measurements ($S^2MP$, $CoperSSM$) need improvements before reaching performances comparable to those used with the HR merged or CR datasets. However, the $S^2MP$ datasets still have two advantages over $CoperSSM$. They currently show better performances against in-situ measurements and are able to provide SM estimates outside of Europe. Hence, it would be interesting to extend the $S^2MP$ algorithm to all the types of land cover and to provide SM estimates over the entire Globe. The objective would be to produce the first Sentinel-only SM dataset available at global scale and to perform deeper comparisons against $SMAPS1$ over large areas. However, the estimation of SM using the $S^2MP$ algorithm outside of Europe would remain challenging due to the in-homogeneous spatial and temporal coverage of S1 (Bauer-Marschallinger et al., 2019).

## 6  Conclusions

The goal of this study was to adapt the $S^2MP$ approach, originally designed to retrieve SM at 10-m resolution over agricultural fields, to a 1-km resolution, which allows to replace S2 by S3 and to significantly improve the NDVI temporal sampling. In addition, the approach was extended to herbaceous land cover areas and tested in six regions over four continents to assess its performances beyond previous evaluations in Southern France.

A very good agreement was found between the S1+S3 and the S1+S2 $S^2MP$ maps for the 6 regions ($R \geq 0.9$, bias $\leq 0.04$ $m^3 m^{-3}$, $STDD \leq 0.03\ m^3 m^{-3}$) meaning that it is possible to replace S2 by S3 NDVI.

The $S^2MP$ maps were then compared to those of the 1-km surface SM product provided by CGLS ($CoperSSM$), which is also a Sentinel-only based dataset. In contrast to $S^2MP$, $CoperSSM$ provides local indices of SM variations. For many applications, they have to be scaled against a reference to transform the variations to actual SM in volumetric units ($m^3 m^{-3}$) before being used. Then, $S^2MP$ was also compared to two HR merged datasets: (*i*) the SWI dataset from CGLS combining S1 and ASCAT measurements ($CoperSWI$) as well as (*ii*) the SMAP+S1 dataset. As for the surface SM dataset, the SWI data had to be scaled into absolute SM values. CGLS products only provide estimates over the European continent and the Mediterranean basin.

Overall, the $S^2MP$ dataset is better correlated to the 1-km surface SM product provided by CGLS over the 4 regions of study in Europe ($R \sim 0.7\text{-}0.8$). Over almost all the pixels within the 6 regions, the $STDD$ between $S^2MP$ and the other HR datasets are lower than 0.06 $m^3 m^{-3}$. In addition, the bias differ significantly inside a same region and can be strongly dry or wet ($\pm$ 0.1 $m^3 m^{-3}$). The correlations between $S^2MP$ and the other HR datasets improve over croplands when the 1-km pixels become homogeneous but a similar behaviour was not clearly found for the other metrics ($STDD$ and bias) and over pixels where the dominant land cover class is herbaceous vegetation.

The $S^2MP$ datasets were also evaluated with respect to in-situ measurements along with the 3 other HR datasets as well as with coarser resolution datasets from SMOS, SMAP and ESA CCI. The coarse resolution (CR) products show higher correlations ($0.68 \leq R \leq 0.77$) than the HR datasets ($0.54 \leq R \leq 0.74$), and the HR merged datasets showed higher correlations than the HR Sentinel-only ones. $S^2MP_{S1S2}$ showed the highest bias with respect to in-situ measurements with -0.07 $m^3 m^{-3}$. Finally, the $STDD$ differ according to the dataset as well as the spatial resolution and range from 0.05 (with $S^2MP_{S1S2}$ for example) to 0.08 $m^3 m^{-3}$ ($CoperSWI$).

In general, these results show that the HR datasets only based on Sentinel ($S^2MP$ and $CoperSSM$) are not as competitive as the other HR merged and CR datasets with respect to in-situ measurements in terms of correlation and bias, but $S^2MP$ still presents several advantages. In contrast to the HR data from Copernicus, the $S^2MP$ data do not depend on auxiliary data to be scaled into volumetric units and can provide SM estimates outside of Europe. $S^2MP_{S1S2}$ also shows lower $STDD$ with respect to in-situ measurements than the Copernicus and the SMAP+S1 datasets. It would be interesting to extend the $S^2MP$ to all the types of land cover and to provide SM maps at the globe scale. $S^2MP_{S1S2}$ would be the first global SM dataset at 1-km resolution only based on Sentinel measurements. It would also allow to perform deeper comparisons against SMAP+S1 over large areas and $S^2MP_{S1S2}$ could be used to assess climate impact at regional level in the future. However, a remaining challenge is to provide HR SM data with comparable spatio-temporal coverage and retrieval quality across different land cover types than those of the state-of-the-art coarse resolution products, such as the SMOS, SMAP and ESA CCI products.

*Author contributions.* RM, NJR-F, NB and MZ designed the study. RM and NJR-F undertook the different evaluations and wrote the first version of the manuscript. HB and NB produced the $S^2MP_{S1S2}$ and $S^2MP_{S1S3}$ maps. CA, NB, WD and MZ participated in the analysis of the results. All the authors contributed to the final version of the manuscript.

*Competing interests.* The authors declare no conflict of interest.

*Acknowledgements.* This research made use of data from the Centre Aval de Traitement des Données SMOS (CATDS) operated for the Centre National d'Etudes Spatiales (CNES) by The Institut Français de Recherche pour l'Exploitation de la Mer (IFREMER) in France, as well as from the Copernicus Global Land Service (CGLS), the National Snow and Ice Data Center (NSIDC), the ESA's Climate Change Initiative for Soil Moisture project, the Centre d'Etudes Spatiales de la BIOsphère (CESBIO), the Institut National Agronomique de Tunisie (INAT) and the International Soil Moisture Network (ISMN). The authors acknowledge partial funding from the ESA's Climate Change Initiative for Soil Moisture project (Contract No. 4000104814/11/I-NB and 4000112226/14/I-NB). RM and NJR-F acknowledge partial funding by the Centre National d'Études Spatiales (CNES) APR TOSCA project SMOS-TE.

**Table 1.** In-situ measurements that were used in this study. The depths are quoted as two numbers: the first one is the upper depth, and the second one is the lower depth of the sensor. Both numbers are equal when the sensor is placed horizontally. The fourth column gives the number of sensors that provide SM measurements in 2019. These measurements were used to convert the relative indices from $CoperSSM$ and $CoperSWI$ into SM estimates with volumetric units (m$^3$ m$^{-3}$, Section 3.2). The number in parenthesis corresponds to the number of in-situ locations where the evaluations of the remotely sensed data were significant (P-value below 5%, Section 3.3).

| Measurements | Location | Depth (m) | Sensors | Reference |
|---|---|---|---|---|
| REMEDHUS | Spain | 0–0.05 | 19 (13) | Gonzalez-Zamora et al. (2018) |
| SMOSMANIA | Southwest of France | 0.05–0.05 | 4 (3) | Calvet et al. (2007) |
| SMOSMANIA | Southeast of France | 0.05–0.05 | 5 (0) | Calvet et al. (2007) |
| OZNET | Australia | 0–0.05 | 11 (10) | Smith et al. (2012); Young et al. (2008) |
| USCRN | North America | 0.05–0.05 | 2 (1) | Bell et al. (2013) |
| ARM | North America | 0.05–0.05 | 24 (13) | Cook (2016, 2018) |
| MERGUELLIL | Tunisia | 0–0.05 | 5 (2) | Amri et al. (2011); Gorrab et al. (2015) |

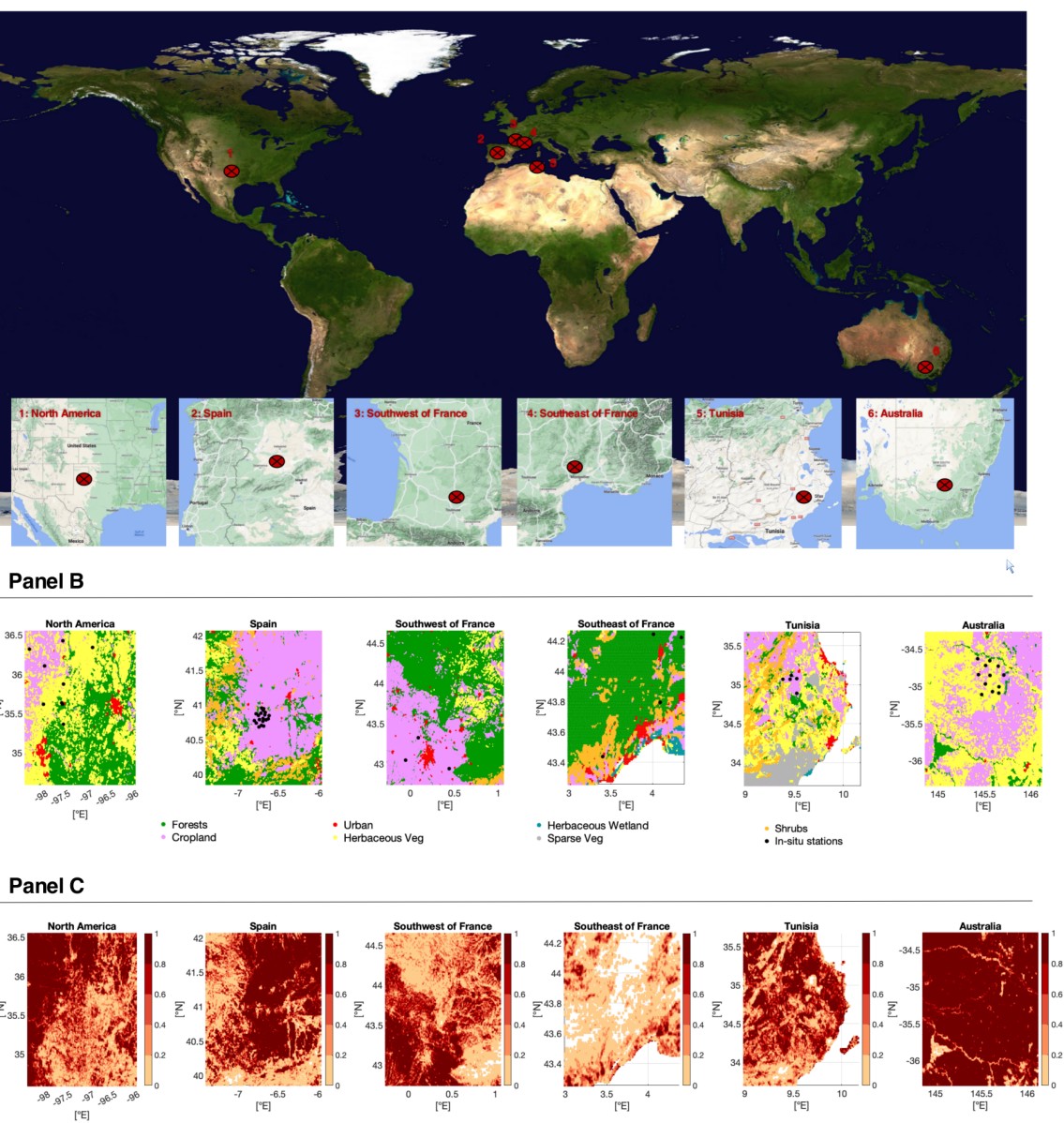

**Figure 1. Panel A:** Global locations of the 6 regions of study. **Panel B:** Copernicus land cover maps of the 6 regions of study aggregated at 1-km spatial resolution. Only the dominant land cover type within a 1-km$^2$ pixel is shown. For instance, a pixel characterised as forests can contain 27% of forests, 26% of croplands, 24% of herbaceous vegetation and 23% of shrublands, or 90% of forests and 10% of herbaceous vegetation. The in-situ stations are shown as black dots. One black dot can correspond to several sensors since some of them have the same coordinates. **Panel C:** Proportion of croplands and herbaceous vegetation within each 1-km$^2$ pixel for the 6 regions of study. The proportion is expressed as a percentage ranging from 0 to 1. Pixels with no cropland or herbaceous vegetation at all are shown as white areas.

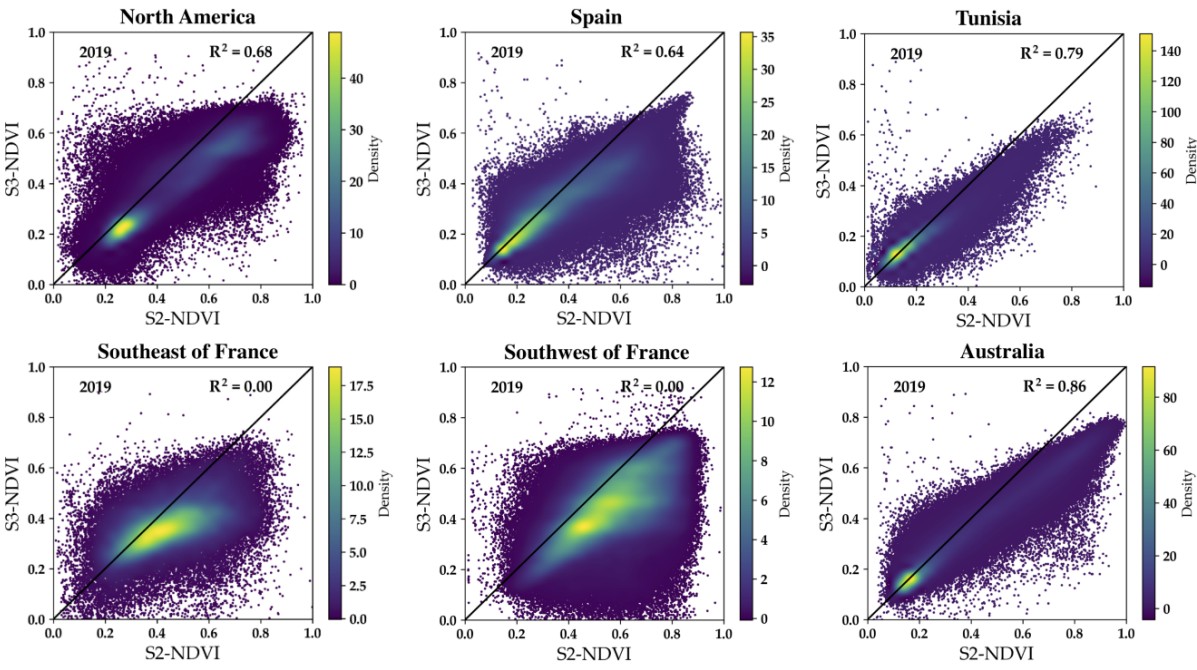

**Figure 2.** Correlation between S2 and S3 NDVI at 1-km grid scale for the 6 study regions from January to December 2019.

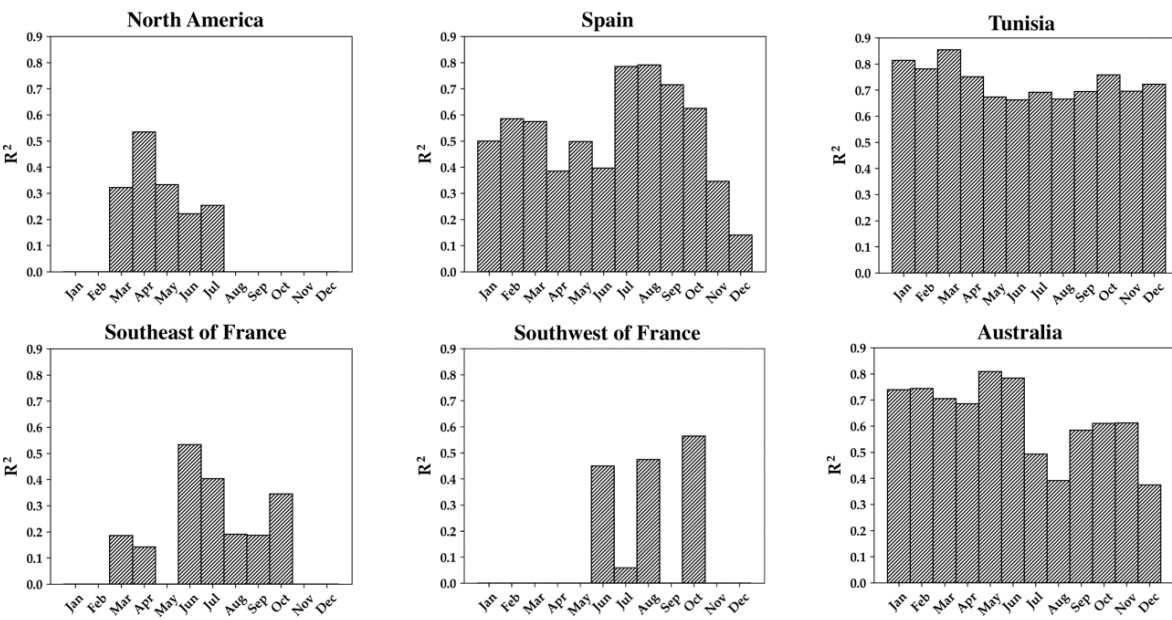

**Figure 3.** Correlation between S2 and S3 NDVI at 1-km grid scale each month for the 6 study regions from January to December 2019. Months having no bars means that there is no correlation between S2 and S3 NDVI.

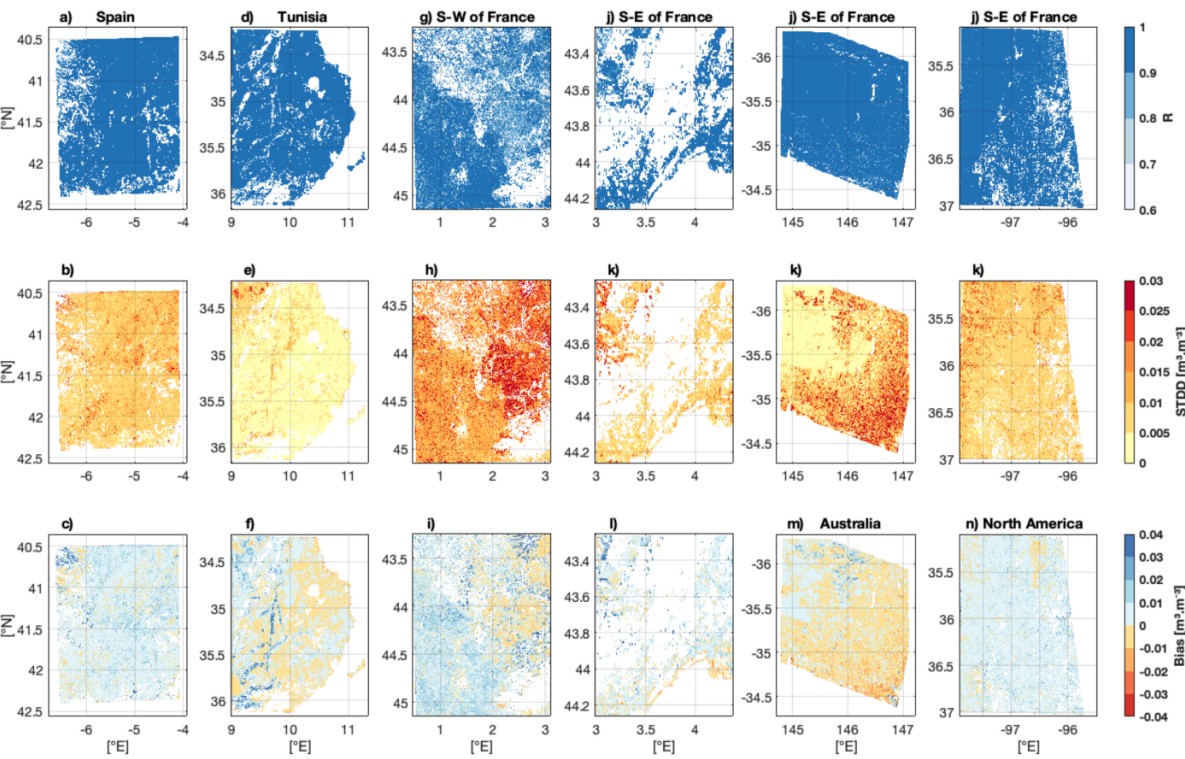

**Figure 4.** Comparison of $S^2MP_{S1S3}$ with respect to $S^2MP_{S1S2}$ over the regions of study in terms of Pearson correlation ($R$) as well as bias ($S^2MP_{S1S2}$ minus $S^2MP_{S1S3}$) and standard deviation of the difference ($STDD$) in $\text{m}^3\,\text{m}^{-3}$. The analysis was performed from January to December 2019.

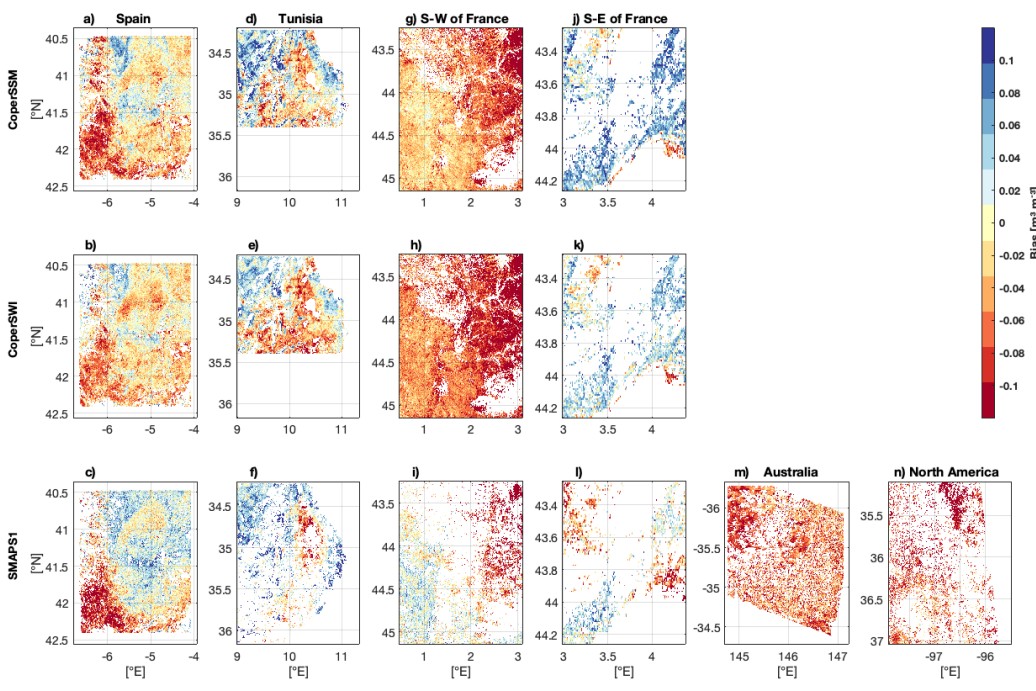

**Figure 5.** Comparison of $S^2MP_{S1S3}$ with respect to $CoperSSM$ ($S^2MP_{S1S3}$ minus $CoperSSM$), $CoperSWI$ ($S^2MP_{S1S3}$ minus $CoperSWI$) and $SMAPS1$ ($S^2MP_{S1S3}$ minus $SMAPS1$) over the regions of study in terms of bias in $\mathrm{m^3\,m^{-3}}$. The analysis was performed from January to December 2019.

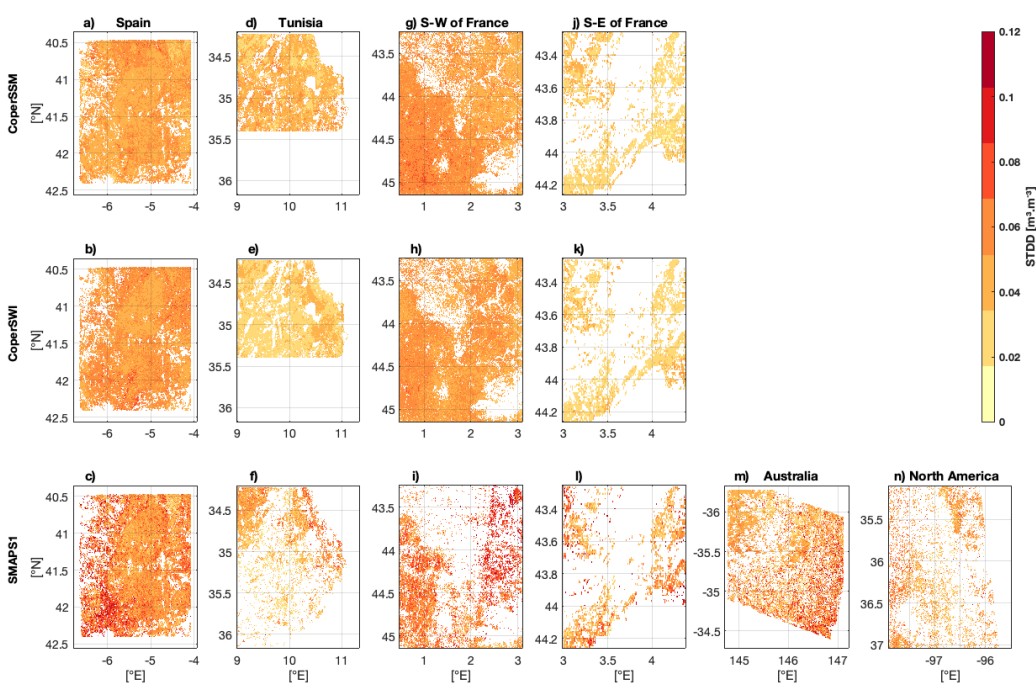

**Figure 6.** Comparison of $S^2MP_{S1S3}$ with respect to $CoperSSM$, $CoperSWI$ and $SMAPS1$ over the regions of study in terms of standard deviation of the difference ($STDD$) in $m^3\,m^{-3}$. The analysis was performed from January to December 2019.

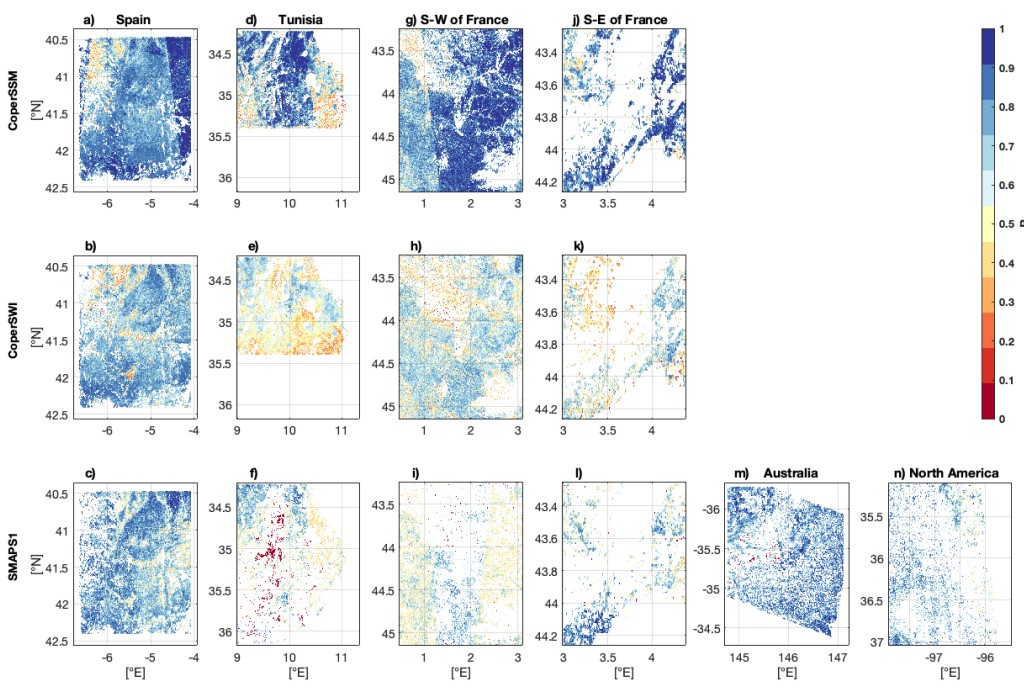

**Figure 7.** Comparison of $S^2MP_{S1S3}$ with respect to $CoperSSM$, $CoperSWI$ and $SMAPS1$ over the regions of study in terms of Pearson correlation ($R$). The analysis was performed from January to December 2019.

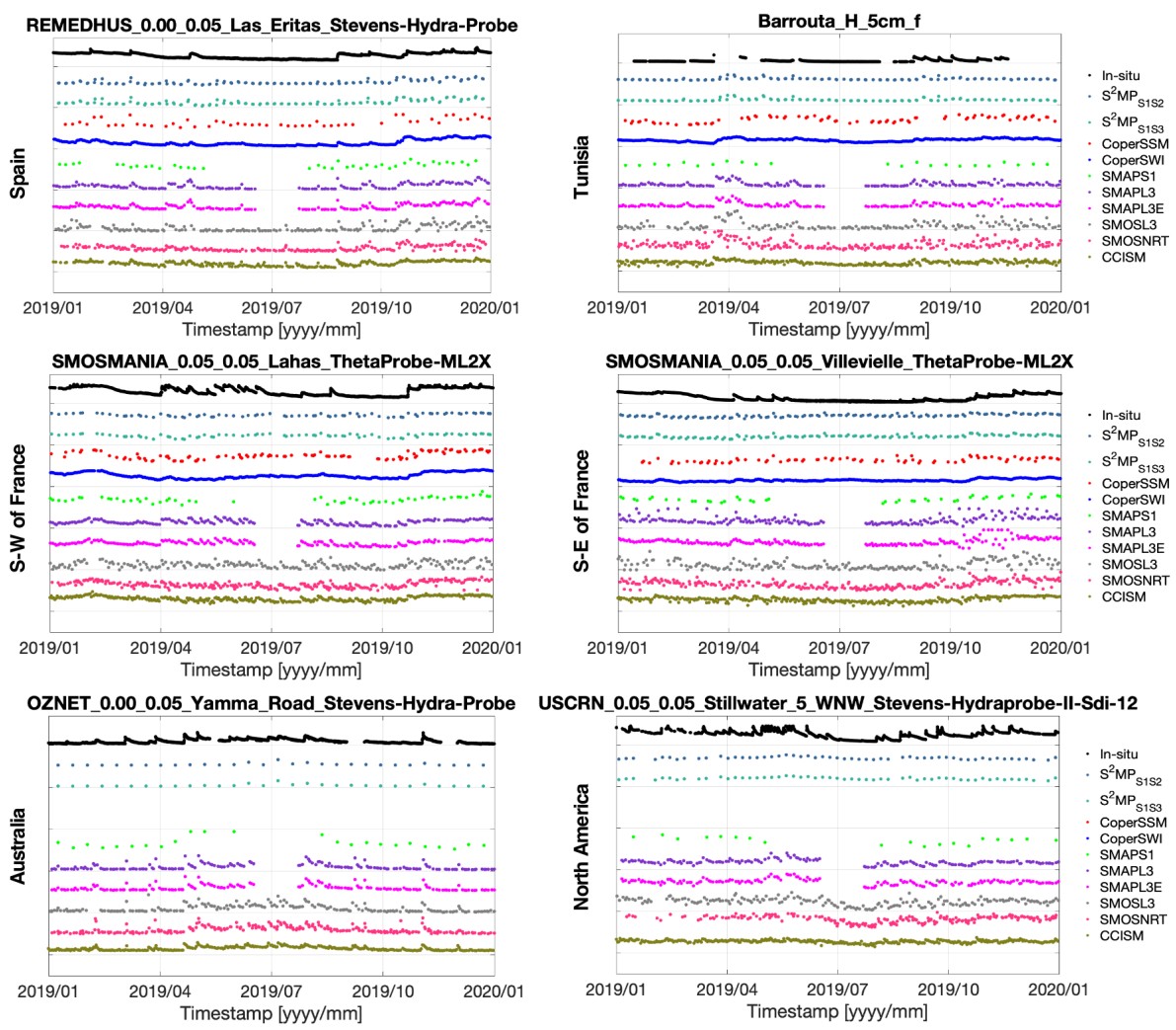

**Figure 8.** Examples of SM time series from the different HR and CR datasets at 6 in-situ stations (one for each region).

**Table 2.** Comparison of $S^2MP_{S1S3}$ against $CoperSSM$, $CoperSWI$ and $SMAPS1$ , in terms of $R$, bias and $STDD$, over 1-km$^2$ pixels where croplands or herbaceous vegetation are the dominant land cover classes, respectively. The metrics are also derived according to the degree of coverage of the land cover. One set of metrics is computed considering only pixels covered by less than 75% of croplands or herbaceous vegetation. Another set of metrics is computed considering only pixels covered by at least 75% of croplands or herbaceous vegetation. The analysis was performed from January to December 2019.

| Regions | Products | Croplands | | | | | | Herbaceous vegetation | | | | | |
|---|---|---|---|---|---|---|---|---|---|---|---|---|---|
| | | $< 75\%$ | | | $\geq 75\%$ | | | $< 75\%$ | | | $\geq 75\%$ | | |
| | | $R$ | $Bias$ | $STDD$ | $R$ | $Bias$ | $STDD$ | $R$ | $Bias$ | $STDD$ | $R$ | $Bias$ | $STDD$ |
| Spain | $CoperSSM$ | 0.63 | 0.01 | 0.06 | 0.72 | 0.01 | 0.06 | 0.67 | 0.06 | 0.06 | 0.69 | 0.07 | 0.06 |
| | $CoperSWI$ | 0.61 | 0.01 | 0.06 | 0.68 | 0.01 | 0.06 | 0.65 | 0.04 | 0.06 | 0.70 | 0.06 | 0.06 |
| | $SMAPS1$ | 0.54 | -0.01 | 0.07 | 0.65 | -0.01 | 0.06 | 0.64 | 0.06 | 0.08 | 0.71 | 0.09 | 0.08 |
| Tunisia | $CoperSSM$ | 0.37 | -0.02 | 0.06 | 0.55 | -0.01 | 0.05 | 0.32 | -0.01 | 0.06 | 0.31 | -0.01 | 0.06 |
| | $CoperSWI$ | 0.37 | 0 | 0.05 | 0.45 | -0.01 | 0.05 | 0.36 | 0.01 | 0.05 | 0.34 | 0.01 | 0.05 |
| | $SMAPS1$ | 0.38 | -0.03 | 0.07 | 0.51 | -0.03 | 0.06 | 0.32 | -0.02 | 0.07 | 0.27 | -0.01 | 0.07 |
| Southwest of France | $CoperSSM$ | 0.56 | 0.03 | 0.06 | 0.71 | 0.03 | 0.06 | 0.69 | 0.07 | 0.06 | 0.62 | 0.09 | 0.06 |
| | $CoperSWI$ | 0.48 | 0.06 | 0.06 | 0.59 | 0.05 | 0.06 | 0.55 | 0.09 | 0.06 | 0.58 | 0.09 | 0.06 |
| | $SMAPS1$ | 0.28 | 0.01 | 0.09 | 0.48 | -0.01 | 0.07 | 0.36 | 0.09 | 0.10 | 0.34 | 0.11 | 0.10 |
| Southeast of France | $CoperSSM$ | 0.63 | -0.04 | 0.04 | 0.75 | -0.05 | 0.03 | 0.44 | 0.02 | 0.05 | 0.72 | 0.06 | 0.02 |
| | $CoperSWI$ | 0.47 | -0.03 | 0.04 | 0.56 | -0.03 | 0.04 | 0.48 | 0.06 | 0.05 | - | - | - |
| | $SMAPS1$ | 0.45 | 0.02 | 0.07 | 0.45 | 0.04 | 0.08 | 0.49 | 0.04 | 0.05 | - | - | - |
| Australia | $CoperSSM$ | - | - | - | - | - | - | - | - | - | - | - | - |
| | $CoperSWI$ | - | - | - | - | - | - | - | - | - | - | - | - |
| | $SMAPS1$ | 0.56 | 0.05 | 0.06 | 0.59 | 0.05 | 0.06 | 0.55 | 0.05 | 0.06 | 0.60 | 0.06 | 0.05 |
| North America | $CoperSSM$ | - | - | - | - | - | - | - | - | - | - | - | - |
| | $CoperSWI$ | - | - | - | - | - | - | - | - | - | - | - | - |
| | $SMAPS1$ | 0.68 | 0.07 | 0.06 | 0.71 | 0.08 | 0.06 | 0.58 | 0.06 | 0.06 | 0.53 | 0.09 | 0.06 |

**Table 3.** Evaluation of the HR and CR SM time series against in-situ measurements in terms of Pearson correlation ($R$,$R^a$), bias (remotely sensed minus ground based SM in [$m^3\,m^{-3}$]) and standard deviation of the difference ($STDD$ in [$m^3\,m^{-3}$]). The metrics were computed by taking into account the 6 regions of study together and only the median values are shown here. The symbol $^*$ indicates the HR datasets averaged at 25-km resolution.The analysis was performed from January to December 2019.

| Products | $R$ | $R^a$ | $Bias$ | $STDD$ |
|---|---|---|---|---|
| Sentinel-only high resolution data | | | | |
| $S^2MP_{S1S2}$ | 0.59 | 0.36 | -0.06 | 0.05 |
| $S^2MP_{S1S3}$ | 0.56 | 0.37 | -0.06 | 0.06 |
| $CoperSSM$ | 0.53 | 0.18 | 0.04 | 0.08 |
| Merged high resolution data | | | | |
| $CoperSWI$ | 0.74 | 0.46 | 0.05 | 0.05 |
| $SMAPS1$ | 0.64 | 0.35 | -0.03 | 0.06 |
| Coarse resolution data | | | | |
| $SMAPL3$ | 0.76 | 0.58 | -0.04 | 0.05 |
| $SMAPL3E$ | 0.77 | 0.59 | -0.04 | 0.05 |
| $SMOSL3$ | 0.67 | 0.47 | -0.03 | 0.07 |
| $SMOSNRT$ | 0.68 | 0.46 | -0.03 | 0.05 |
| $CCISM$ | 0.71 | 0.50 | 0.03 | 0.05 |
| High resolution data aggregated to coarse resolution | | | | |
| $S^2MP^*_{S1S2}$ | 0.58 | 0.38 | -0.06 | 0.06 |
| $S^2MP^*_{S1S3}$ | 0.56 | 0.38 | -0.05 | 0.06 |
| $CoperSSM^*$ | 0.53 | 0.20 | 0.05 | 0.07 |
| $CoperSWI^*$ | 0.73 | 0.47 | 0.05 | 0.05 |
| $SMAPS1^*$ | 0.79 | 0.44 | -0.02 | 0.04 |

**Table 4.** Evaluation of the HR and CR SM time series against in-situ measurements in terms of Pearson correlation ($R$,$R^a$), bias (remotely sensed minus ground based SM in [m$^3$ m$^{-3}$]) and standard deviation of the difference ($STDD$ in [m$^3$ m$^{-3}$]). The metrics were computed by taking into account the 6 regions of study together and only the median values are shown here. The symbol $^*$ indicates the HR datasets averaged at 25-km resolution. The analysis was performed from January to December 2019.

| Products | $R$ | $R^a$ | $Bias$ | $STDD$ |
|---|---|---|---|---|
| Sentinel-only high resolution data | | | | |
| $S^2MP_{S1S2}$ | 0.59 | 0.36 | -0.06 | 0.05 |
| $S^2MP_{S1S3}$ | 0.56 | 0.37 | -0.06 | 0.06 |
| $CoperSSM$ | 0.53 | 0.18 | 0.04 | 0.08 |
| Merged high resolution data | | | | |
| $CoperSWI$ | 0.74 | 0.46 | 0.05 | 0.05 |
| $SMAPS1$ | 0.64 | 0.35 | -0.03 | 0.06 |
| Coarse resolution data | | | | |
| $SMAPL3$ | 0.81 | 0.55 | -0.05 | 0.05 |
| $SMAPL3E$ | 0.81 | 0.52 | -0.05 | 0.05 |
| $SMOSL3$ | 0.69 | 0.49 | -0.03 | 0.06 |
| $SMOSNRT$ | 0.76 | 0.45 | -0.02 | 0.05 |
| $CCISM$ | 0.73 | 0.48 | 0.03 | 0.05 |
| High resolution data aggregated to coarse resolution | | | | |
| $S^2MP^*_{S1S2}$ | 0.58 | 0.38 | -0.06 | 0.06 |
| $S^2MP^*_{S1S3}$ | 0.56 | 0.38 | -0.05 | 0.06 |
| $CoperSSM^*$ | 0.53 | 0.20 | 0.05 | 0.07 |
| $CoperSWI^*$ | 0.73 | 0.47 | 0.05 | 0.05 |
| $SMAPS1^*$ | 0.79 | 0.44 | -0.02 | 0.04 |

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
