# Peer review of "Soil moisture estimates at 1-km resolution making a synergistic use of Sentinel data"

_EGUsphere, 2022_

## Referee Comment (RC1)

**"Soil moisture retrieval at 1-km resolution making a synergistic use of Sentinel-1/2/3 data"**
**by Madelon, R. et al.**

**General comment:**

This study aims at producing and evaluating a new soil moisture dataset at 1km spatial resolution, based on both SAR backscatter data and the normalized vegetation index (NDVI) from the Sentinel missions. To reach their objective the Authors adapted the $S^2MP$ algorithm, developed in El Hajj et al. (2017), to move from plot scale to 1km spatial resolutions. To reach the target resolution the Authors investigate advantages on using NDVI from Sentinel-3 (S3) instead of Sentinel-2 (S2; higher resolution as compared to S3), used in previous works (El Hajj et al., 2017). This work is of interest for the journal. Despite this there are major gaps that should be filled. For instance, it would be helpful to elaborate more on the methodology and the Sentinel datasets used (i.e., pre-processing) as well as on the final objective of this study. The introduction section and the abstract need to be revised, focusing on the main target of this work. Finally, the title seems to be not appropriate for the analysis conducted.
Specific comments can be found below.

**Specific comments:**

**RC1-Title:** The Authors should consider another title. The actual version is misleading, suggesting that three products (S1, S2 and S3) are used together to derive the 1km soil moisture dataset, which is not the case considering that the algorithm only uses two of them.

**RC2-L74:** Which is the main reason for using S3 dataset (i.e., NDVI) as compared to S2? Which is (or which should be) the major advantage? Which is the difference between the sensors? This aspect is not treated in detail. A description of the Sentinel missions (not only Sentinel-1) should be included, to better understand which is the usefulness of S3 (i.e., in terms of spatial and temporal resolution and/or processing computation timing). The usefulness of S3 in terms of temporal sampling is only mentioned at line 422. Clarifying this aspect would be useful to better frame the analysis.

**RC3-L70-85:** I struggle to see the final objectives of this work. Want the Authors to demonstrate which NDVI product is more advantageous between S2 and S3? Do the Authors think that in the future they will move to S3 NDVI? In theory, if good agreement was found between S1+S2 and S1 +S3 (line 425), why not using S3 for the following analysis? I think the Authors should elaborate more on their objective and provide insights on what pushed them to use S3 and later on staying with the initial S2 product. This should be clarified in both the introduction section and in the abstract.

**RC4–Section2:** I did not find a proper description of the Sentinel products used to run the $S^2MP$ algorithm (S1, S2 and S3). The $S^2MP$ section 2.1.1 should go in the method while a sub-section to describe the Sentinel datasets used in $S^2MP$ (i.e., original spatial and temporal resolution) should be added. As an additional suggestion, the HR and CR products used for evaluation could go in another sub-section (i.e., Satellite products used for evaluation) reducing the detailed description and directly referring the reader to the reference work. For instance, at section 2.1.2 both the 1-km and 3-km SMAP+S1 L2 products are described but only the 1-km product is then used.

**RC5–L106:** SMAP provides passive measurements of *brightness temperature* (passive sensor) in vertical and horizontal polarization and not measurements of land surface SM. This part should be edited.

**RC6–L178:** *"The Dynamic Land Cover Map product ... provided by CGLS was used to evaluate the*

*different HR and CR data sets"*? In my understanding the objective should be to evaluate the new S$^2$MP product and not the other HR and CR SM products. Please comment on that.

**RC7–Figure 1:** It's difficult to locate the different test sites. I suggest to add an inset in Figure1 summarizing where the test sites are located. Additionally, it would be useful to add in the maps the in-situ soil moisture sensors location (this would help also the discussion section – line 386).

**RC8–L194-195:** This part was already mentioned in Section–2.1.1. I suggest to merge the two sections and, as in my previous comment (RC4), to describe the Sentinel products in the data section instead of the S$^2$MP algorithm.

**RC9–L201-208:** It is only mentioned that the Authors used S3 at 1km spatial resolution while that S1+S2 SM is obtained moving from a 100 m resolution to a 1km spatial resolution. S3 images should have an original spatial resolution of 300m (please check at line 208). How were the images processed? (If a specific product was used it should be cited). Does the S3 NDVI take into account different land uses as the S2 NDVI product? In any case it could be useful to have an idea of the average percentage of crop and herbaceous vegetation for each product over the study areas. Please comment on that.

**RC10–L203:** It is mentioned that only croplands and herbaceous vegetation are used to derive $S^2MP_{S1S2}$. However, in Figure 1 (land cover processed at 1km spatial scale) over some specific regions (i.e., Southwest and Southeast France or North America), forests are the dominant land use. If the Authors are focusing on cropland and herbaceous vegetation, I am wondering why those areas where not masked (i.e., line 331-332 "*As discussed above, the correlation maps show some features related to the dominant land cover class, in particular, higher correlations are found for areas dominated by croplands and herbaceous vegetation*"). Please comment on that.

**RC11–L218:** Why did the Authors select a threshold of 5%? Please comment on that.

**RC12–L212:** Why only ascending orbits from SMOS and descending orbits from SMAP were used? It is not explained in the text or at least I did not find it. Please comment on that.

**RC13–L242-244:** "*The S2 NDVI at 1 km grid ... and compared to the S3 NDVI obtained at 1km*". This sentence is clearly part of the method. I suggest to remove it from the results section.

**RC14–L273-274:** Considering the original spatial resolution of S3 an explanation on how this product was processed would be very helpful to understand the results. See my previous comment RC9.

**RC15–L276:** "*However, taking into account the overall very good agreement of $S^2MP_{S1S2}$ and $S^2MP_{S1S3}$ maps, for the sake of simplicity and clarity, in the following sections only $S^2MP_{S1S2}$ is compared to the other HR data sets*". This choice makes it difficult to understand the objective of the paper. Which is the point to use the S3 product? This aspect should be clarified throughout the text.

**RC16–L369:** The Authors should edit the first line of the discussion section. It suggests that four HR soil moisture dataset were produced and evaluated. Whereas, the product evaluated here should be S$^2$MP against the other data sets.

**RC17–Section 4.4:** The manuscript does not show any time series analysis. It could be interesting to see some time series of the S$^2$MP product against in situ data and the other HR-CR soil moisture products.

**RC18:** The Authors should check for the acronyms in the abstract section as well as throughout the text. The notation should be uniform. For instance, at line 5 the Authors write NDVI (Normalized Different Vegetation Index) but it should be the opposite considering other notations (i.e. Soil Moisture [SM]). Another example is at line 194 where the notation "soil moisture" should be edited to "SM".

**RC19–L130:** Maybe the Authors can mention that T is the "*characteristic time length*".

**RC20-L118:** The term *"estimation"* before images should be removed.

**REFERENCES**

El Hajj, M., Baghdadi, N., Zribi, M., and Bazzi, H.: Synergic use of Sentinel-1 and Sentinel-2 images for operational soil moisture mapping at high spatial resolution over agricultural areas., Remote Sensing, 9, 1292, https://doi.org/doi:10.3390/rs9121292, 2017.

---

## Author Comment (AC1)

**General comment:**
This study aims at producing and evaluating a new soil moisture dataset at 1 km spatial resolution, based on both SAR backscatter data and the normalized vegetation index (NDVI) from the Sentinel missions. To reach their objective the Authors adapted the S2MP algorithm, developed in El Hajj et al. (2017), to move from plot scale to 1 km spatial resolutions. To reach the target resolution the Authors investigate advantages on using NDVI from Sentinel-3 (S3) instead of Sentinel-2 (S2; higher resolution as compared to S3), used in previous works (El Hajj et al., 2017). This work is of interest for the journal. Despite this there are major gaps that should be filled. For instance, it would be helpful to elaborate more on the methodology and the Sentinel datasets used (i.e., pre-processing) as well as on the final objective of this study. The introduction section and the abstract need to be revised, focusing on the main target of this work. Finally, the title seems to be not appropriate for the analysis conducted.
Specific comments can be found below.

We truly thank the reviewer for his/her pertinent comments on the manuscript. We understand that the objective of this work was not clear enough in the current version. Retrieving soil moisture is currently done globally at low spatial resolution (40 km) or over smaller regions at high (~1 km) or very high spatial resolution (10-100 m) as the original S$^2$MP algorithm does over croplands. The goal of this manuscript is to study how the S$^2$MP approach could be used to produce 1-km soil moisture maps over large regions using the synergies of different Sentinel satellites. Of course, the first step is to be able to use it over larger regions and to extend S$^2$MP to other land cover types than croplands. In addition, since for soil moisture mapping at large scale, 1 km is a high spatial resolution, using S3 instead of S2 becomes possible. In spite of the lower spatial resolution of S3, its higher revisit frequency with respect to S2 makes it possible to reduce uncertainties in the NDVI estimations used by S$^2$MP introduced by clouds and gaps in the optical time series.
In this context, we understand that comparing the S1+S2 vs S1+S3 results and continuing showing the performances of S1+S2 against those of other state-of-the-art datasets was a bad choice. In a revised version of the manuscript we will focus on the results for S1+S3 in addition to clarify the goal in the abstract and introduction as well as improve the structure and content of the data and method sections.

**Specific comments:**

**RC1-Title:** The Authors should consider another title. The actual version is misleading, suggesting that three products (S1, S2 and S3) are used together to derive the 1km soil moisture dataset, which is not the case considering that the algorithm only uses two of them.

The title will be turned into "Soil moisture estimates at 1-km resolution making a synergistic use of Sentinel data".

**RC2-L74:** Which is the main reason for using S3 dataset (i.e., NDVI) as compared to S2? Which is (or which should be) the major advantage? Which is the difference between the sensors? This aspect is not treated in detail. A description of the Sentinel

missions (not only Sentinel-1) should be included, to better understand which is the usefulness of S3 (i.e., in terms of spatial and temporal resolution and/or processing computation timing). The usefulness of S3 in terms of temporal sampling is only mentioned at line 422. Clarifying this aspect would be useful to better frame the analysis.

Since 1 km spatial resolution is a high spatial resolution for large scale mapping of soil moisture, the full spatial resolution of S1 and S2 is not needed. S2 data used as input to the S2MP algorithm is NDVI. Therefore, for this target resolution, S2 could be replaced by S3.

The higher temporal revisit of S3 allows to retrieve more optical images without cloud conditions than those onboard S2. This results in a better estimation of the vegetation effect on the SM retrieval through a more precise NDVI computation.

However, the counterback of using S3 with respect to S2 is that the NDVI used comes from the 1 km^2 pixels while the backscattering from S1 will come only from croplands and herbaceous regions within the 1 km^2 pixel. The possible uncertainties introduced by this mismatch have to be evaluated. It is shown in the manuscript that actual differences in the output soil moisture estimates are low. This was somehow expected since the main predictor for soil moisture remains the microwave backscattering.

Of course, producing SM daily maps at 1 km resolution using S3 instead of S2 is easier and faster because there is no need to do all the previous preprocessing of S2 to analyze and aggregate the data from its original resolution to 1 km.

We also agree that information about the S2 and S3 missions are lacking in the manuscript. Please find below a full description of the Sentinel missions and the data used for the production of the $S^2MP_{S1S2}$ and $S^2MP_{S1S3}$ SM maps.

- The Sentinel-1 mission is the first satellite constellation mission of the Copernicus program and was conducted by ESA. The mission is composed of a constellation of two satellites sharing the same orbital plane. S1A was launched on 3 April 2014, and S1B on 25 April 2016. They were placed in a near-polar, sun-synchronous orbit. The revisit frequency is 12 days (6 days using both satellites) with crossing time at equator at 6:00 pm for the descending overpass. S1A and S1B carry onboard a C-band (wavelength 6 cm) SAR imaging instrument, enable to acquire imagery regardless of the weather and the time of the day. Four imaging modes with different resolution (down to 5 m) and coverage (up to 400 km) are available with this instrument, which allow a continuous radar mapping of the Earth. S1B has now retired.
  For the production of the $S^2MP$ SM maps, S1A and S1B SAR images were collected over each region of study. S1 images are accessible from the Copernicus website (https://scihub.copernicus.eu/dhus//home). The S1 images (10 m x 10 m) were acquired in the Interferometric Wide-swath (IW) imagining mode with VV and VH polarizations and the S1 Toolbox (S1TBX) developed by ESA was used to calibrate the images. This calibration aims to

convert digital number values from S1 images into backscattering coefficients in a linear unit and ortho-rectifying the images using the Shuttle Radar Topography Mission (SRTM) Digital Elevation Model (DEM).

- The Sentinel-2 mission aims at monitoring variability in land surface conditions and was developed by ESA in the framework of the Copernicus program. S2A and S2B were launched on 23 June 2015 and 7 March 2017, and were placed in a near polar, sun-synchronous orbit. The revisit frequency is 10 days (5 days with 2 satellites) and the descending orbit crossing time at equator is at 10:30 am. The spatial coverage ranges from 56° S to 84° N. The satellites carry onboard a multi-spectral instrument with 13 bands: 4 bands at 10-m spatial resolution, 6 bands at 20-m and 3 bands at 60-m spatial resolution. The orbital swath width is 290 km.
  For the production of the $S^2MP$ SM maps based on S1 and S2, optical images from S2A on dates close to S1 SAR images (less than 2 weeks) were downloaded from the French land data service center (Theia) website (https://www.theia-land.fr/). The S2A optical images (10 m x 10 m) are corrected for atmospheric effects and ortho-rectified.

- The Sentinel-3 satellites were launched by ESA to answer the operational needs of the Copernicus program by measuring sea-surface topography, sea- and land-surface temperature, ocean and land color with high-end accuracy and reliability. The main objectives are to support ocean forecasting systems as well as environmental and climate monitoring. S3A and S3B were launched on 16 February 2016 and on 25 April 2018, respectively. The S3 satellites orbit is a near-polar, sun-synchronous orbit with crossing time at equator at 10:00 am for the descending overpass. They carry onboard an optical instrument payload, the Ocean and Land Color Instrument (OLCI), that samples 21 spectral bands ([0.4-1.02] μm) with a swath width of 1 270 km and a spatial resolution of 300 m. They also carry a dual-view scanning temperature radiometer at 500-m spatial resolution: the Sea and Land Surface Temperature Radiometer (SLSTR). The revisit frequency of these instruments is 2 days when both satellites are used together.
  In this study, the S3 10-days synthesis NDVI at 1-km spatial resolution were used for the production of the $S^2MP$ SM maps based on S1 and S3. These data are accessible in the SY_2_V10 product and were downloaded from the Copernicus website (https://scihub.copernicus.eu/dhus//home). The data from this product rely upon the synergistic use of the OLCI and SLSTR instruments. The product provides a 1-km VEGETATION-Like dataset including 10-day synthesis surface reflectances and NDVI. The NDVI values correspond to a maximum NDVI value composite of all segments received for 10 days.

We will revise the data and method sections to better present the essential information concerning the Sentinel missions and the data used as inputs to the $S^2MP$ algorithm.

**RC3-L70-85:** I struggle to see the final objectives of this work. Want the Authors to demonstrate which NDVI product is more advantageous between S2 and S3? Do the Authors think that in the future they will move to S3 NDVI? In theory, if good agreement was found between S1+S2 and S1 +S3 (line 425), why not using S3 for the following analysis? I think the Authors should elaborate more on their objective and

provide insights on what pushed them to use S3 and later on staying with the initial S2 product. This should be clarified in both the introduction section and in the abstract.

As said above, we agree with the reviewer. Since a good agreement was found between S1+S2 and S1+S3 we continued showing more results on S1+S2 but we understand that it was a bad choice and that it confused the readers. It would have been a better choice to continue the discussion using the S1+S3 retrievals. The reviewer is right in the goal and more details on the interest of S3 with respect to S2 have already been given in the answer to RC2-L74, we will not repeat them here again.

Please find below figures showing the comparison between the $S^2MP_{S1S3}$ SM maps and those of the other HR datasets in terms of bias, STDD and R.

[Figure]

**Figure 5.** Comparison of $S^2MP_{S1S3}$ with respect to $CoperSSM$ ($S^2MP_{S1S3}$ minus $CoperSSM$), $CoperSWI$ ($S^2MP_{S1S3}$ minus $CoperSWI$) and $SMAPS1$ ($S^2MP_{S1S3}$ minus $SMAPS1$) over the regions of study in terms of bias in $m^3 \, m^{-3}$. The analysis was performed from January to December 2019.

[Figure]

**Figure 6.** Comparison of $S^2MP_{S1S3}$ with respect to $CoperSSM$, $CoperSWI$ and $SMAPS1$ over the regions of study in terms of standard deviation of the difference ($STDD$) in m$^3$ m$^{-3}$. The analysis was performed from January to December 2019.

[Figure]

**Figure 7.** Comparison of $S^2MP_{S1S3}$ with respect to $CoperSSM$, $CoperSWI$ and $SMAPS1$ over the regions of study in terms of Pearson correlation ($R$). The analysis was performed from January to December 2019.

In a revised version of the manuscript we will replace the original Figure 5-7 ($S^2MP_{S1S2}$ maps versus the other HR maps) by these new figures in the revised version of the manuscript. Sections results and discussion will be updated accordingly.

**RC4–Section2:** I did not find a proper description of the Sentinel products used to run the S2 MP algorithm (S1, S2 and S3). The S2 MP section 2.1.1 should go in the method while a subsection to describe the Sentinel datasets used in S2 MP (i.e., original spatial and temporal resolution) should be added. As an additional

suggestion, the HR and CR products used for evaluation could go in another sub-section (i.e., Satellite products used for evaluation) reducing the detailed description and directly referring the reader to the reference work. For instance, at section 2.1.2 both the 1-km and 3-km SMAP+S1 L2 products are described but only the 1-km product is then used.

We understand that there is a lack of information concerning the description of the $S^2MP$ algorithm and the Sentinel data used as inputs. This information can be added to the Data section. Otherwise, the suggestions of the reviewer of moving the S2MP description to Methods and regrouping and reducing the text on the other SM data sets used for the evaluation make fully sense and we will take it into account in a manuscript revision

Please find below a full description of the S1/2/3 data used to produce the $S^2MP$ SM maps.

- For the production of the $S^2MP$ SM maps, S1A and S1B SAR images were collected over each region of study. S1 images are accessible from the Copernicus website (https://scihub.copernicus.eu/dhus//home) and the revisit frequency is 12 days (6 days using both satellites). The S1 images (10 m x 10 m) were acquired in the Interferometric Wide-swath (IW) imagining mode with VV and VH polarizations and the S1 Toolbox (S1TBX) developed by ESA was used to calibrate the images. This calibration aims to convert digital number values from S1 images into backscattering coefficients in a linear unit and ortho-rectifying the images using the Shuttle Radar Topography Mission (SRTM) Digital Elevation Model (DEM).
- For the production of the $S^2MP$ SM maps based on S1 and S2, optical images from S2A on dates close to S1 SAR images (less than 2 weeks) were downloaded from the French land data service center (Theia) website (https://www.theia-land.fr/). The S2A optical images (10 m x 10 m) are corrected for atmospheric effects and ortho-rectified. The revisit frequency is 10 days (5 days with 2 satellites).
- The S3 10-days synthesis NDVI at 1-km spatial resolution were used for the production of the $S^2MP$ SM maps based on S1 and S3. These data are accessible in the SY_2_V10 product and were downloaded from the Copernicus website (https://scihub.copernicus.eu/dhus//home). The data from this product rely upon the synergistic use of the Ocean and Land Color Instrument (OLCI) and Sea and Land Surface Temperature Radiometer (SLSTR. The product provides a 1-km VEGETATION-Like dataset including 10-day synthesis surface reflectances and NDVI. The NDVI values correspond to a maximum NDVI value composite of all segments received for 10 days.The OLCI samples 21 spectral bands ([0.4-1.02] µm) with a swath width of 1 270 km and a spatial resolution of 300 m. The SLSTR works at 500-m resolution. The revisit frequency of these two instruments is 2 days when both satellites are used together.

.

**RC5–L106:** SMAP provides passive measurements of *brightness temperature* (passive sensor) in vertical and horizontal polarization and not measurements of land surface SM. This part should be edited.

This sentence will be corrected.

**RC6–L178:** *"The Dynamic Land Cover Map product … provided by CGLS was used to evaluate the different HR and CR data sets"*? In my understanding the objective should be to evaluate the new S2 MP product and not the other HR and CR SM products. Please comment on that.

The reviewer is fully right. The goal is to evaluate the $S^2MP$ product by comparison to other SM (HR and CR) data sets. The formulation used is misleading and hence, it will be edited and clarified.

**RC7–Figure 1:** It's difficult to locate the different test sites. I suggest to add an inset in Figure 1 summarizing where the test sites are located. Additionally, it would be useful to add in the maps the in situ soil moisture sensors location (this would help also the discussion section – line 386).

Please find below a new version of Figure 1. As advised, it has been modified to help the reader to better locate the different regions of studies. Locations of in-situ stations have also been added. This new figure can replace the original Figure 1 into the manuscript.

[Figure]

**Figure 1. Panel A:** Global locations of the 6 regions of study. **Panel B:** Copernicus land cover maps of the 6 regions of study aggregated at 1-km spatial resolution. Only the dominant land cover type within a 1-km$^2$ pixel is shown. For instance, a pixel characterised as forests can contain 27% of forests, 26% of croplands, 24% of herbaceous vegetation and 23% of shrublands, or 90% of forests and 10% of herbaceous vegetation. The in-situ stations are shown as black dots. One black dot can correspond to several sensors since some of them have the same coordinates. **Panel C:** Proportion of croplands and herbaceous vegetation within each 1-km$^2$ pixel for the 6 regions of study. The proportion is expressed as a percentage ranging from 0 to 1. Pixels with no cropland or herbaceous vegetation at all are shown as white areas.

**RC8–L194-195:** This part was already mentioned in Section 2.1.1. I suggest to merge the two sections and, as in my previous comment (RC4), to describe the Sentinel products in the data section instead of the S2MP algorithm.

We understand that the structure of the data and method sections could be improved. The section data will be revised to only present the description of the Sentinel missions, the Sentinel data used for the production of the S$^2$MP SM maps, and the other HR and CR SM products used in this study. The section method will describe the S$^2$MP algorithm and SM maps, as well as how the evaluations were carried out. Hence, the initial S$^2$MP description from the section data will move to the section method.

**RC9–L201-208:** It is only mentioned that the Authors used S3 at 1 km spatial resolution while that S1+S2 SM is obtained moving from a 100 m resolution to a 1 km spatial resolution. S3 images should have an original spatial resolution of 300m (please check at line 208). How were the images processed? (If a specific product was used it should be cited). Does the S3 NDVI take into account different land uses as the S2 NDVI product? In any case it could be useful to have an idea of the average percentage of crop and herbaceous vegetation for each product over the study areas. Please comment on that.

Regarding the 300m to 1 km question, in this study, the S3 10-days synthesis NDVI at 1-km spatial resolution were used for the production of the $S^2$MP SM maps based on S1 and S3. These data are accessible in the SY_2_V10 product and were downloaded from the Copernicus website (https://scihub.copernicus.eu/dhus//home). The data from this product rely upon the synergistic use of the OLCI and SLSTR instruments. The product provides a 1-km VEGETATION-Like dataset including 10-day synthesis surface reflectances and NDVI. The NDVI values correspond to a maximum NDVI value composite of all segments received for 10 days.
Regarding the different land covers within the 1 km pixels, this is the goal of the S3 vs S2 NDVI comparison in the Results section.

**RC10–L203:** It is mentioned that only croplands and herbaceous vegetation are used to derive *S2 MPS1S2*. However, in Figure 1 (land cover processed at 1 km spatial scale) over some specific regions (i.e., Southwest and Southeast France or North America), forests are the dominant land use. If the Authors are focusing on cropland and herbaceous vegetation, I am wondering why those areas where not masked (i.e., line 331-332 "*As discussed above, the correlation maps show some features related to the dominant land cover class, in particular, higher correlations are found for areas dominated by croplands and herbaceous vegetation*"). Please comment on that.

This work extends the $S^2$MP algorithm including herbaceous regions with respect to the croplands used by El Hajj et al. However, for the evaluation we think it is interesting not to limit ourselves to homogenous regions with only herbaceous and cropland classes. The final goal of course is to have good soil moisture retrievals at high resolution over all types of surfaces. Certainly, we are not there yet but evaluating the performances on mixed land cover pixels with state of the art algorithms helps to understand the work that remains to be done.

**RC11–L218:** Why did the Authors select a threshold of 5%? Please comment on that.

The 2.5% lowest and 2.5% highest values are discarded from the scaling reference, (the in-situ measurements here) to remove the potential outliers that can be caused by instrumental noise. This is a threshold advised in Brocca et al. 2011. The sentence in line 218 is not clear and will be rephrased. Of course, the above reference will be cited into the manuscript.

Brocca, L., Hasenauer, S., Lacava, T., Melone, F., Moramarco, T., Wagner, W., Dorigo, W., Matgen, P., Martínez-Fernández, J., Llorens, P., Latron, J., Martin, C., and M, B.: Soil moisture estimation through ASCAT and AMSR-E sensors: An intercomparison and validation study across Europe, Remote Sensing of Environment, 115, 3390–3408, 201.

**RC12–L212:** Why only ascending orbits from SMOS and descending orbits from SMAP were used? It is not explained in the text or at least I did not find it. Please comment on that.

During the night and early in the morning, the soil skin layer is closer to thermal equilibrium, meaning that the vegetation temperature is closer to the soil temperature. During the afternoon, the balance is lost and the vegetation temperature is closer to the air temperature leading to more uncertain soil moisture retrievals of lower quality. This is often reflected by lower performances against in-situ measurements for the afternoon SM estimates than those of the morning (Leroux et al., 2014). Also, in some regions, convective precipitation in the afternoon makes more complex the soil moisture retrieval with afternoon orbits.
This is the reason why only morning orbits from SMOS (ascending overpasses) and SMAP (descending overpasses) were used. This justification will be added to the manuscript.

**RC13–L242-244:** *"The S2 NDVI at 1 km grid ... and compared to the S3 NDVI obtained at 1km"*. This sentence is clearly part of the method. I suggest to remove it from the results section.

As advised, this sentence will be removed and moved to the section method.

**RC14–L273-274:** Considering the original spatial resolution of S3 an explanation on how this product was processed would be very helpful to understand the results. See my previous comment RC9.

Please see our response to RC9 for more details concerning the description of the S$^2$MP SM data production. As already explained before, the section data of the revised manuscript will present all the different types of Sentinel data used as inputs to the S$^2$MP algorithm and the way how they have been processed will be explained in the section method.

**RC15–L276:** *"However, taking into account the overall very good agreement of S2MPS1S2 and S2MPS1S3 maps, for the sake of simplicity and clarity, in the following sections only S2MPS1S2 is compared to the other HR data sets"*. This choice makes it difficult to understand the objective of the paper. Which is the point to use the S3 product? This aspect should be clarified throughout the text.

As mentioned in RC3, we understand that this was a bad choice and that it would be more logical to pursue the analysis with the S1+S3 SM maps after showing there is a good agreement between the S1+S2 and S1+S3 maps. The results of the comparison

between the S1+S3 SM maps and those of the other HR datasets in terms of bias, STDD and R can be seen in the answer to RC3.
We will replace the original Figure 5-7 (S1+S2 versus the other HR maps) by these new figures in the revised version of the manuscript. Sections results and discussion will be updated accordingly.

**RC16–L369:** The Authors should edit the first line of the discussion section. It suggests that four HR soil moisture dataset were produced and evaluated. Whereas, the product evaluated here should be S2MP against the other data sets.

This sentence will be rephrased.

**RC17–Section 4.4:** The manuscript does not show any time series analysis. It could be interesting to see some time series of the S2MP product against in situ data and the other HR-CR soil moisture products.

Actually, the computation of the different statistical metrics (R, bias, STDD) with respect to the other datasets or the in-situ measurements is  a time series analysis. However, we clearly understand the interest of showing time series from the different datasets. The figure below shows time series from each dataset at 6 in-situ stations. The revised version of the manuscript will include this new figure.

[Figure]

**Figure 8.** Examples of SM time series from the different HR and CR datasets at 6 in-situ stations (one for each region).

**RC18:** The Authors should check for the acronyms in the abstract section as well as throughout the text. The notation should be uniform. For instance, at line 5 the Authors write NDVI (Normalized Difference Vegetation Index) but it should be the opposite considering other notations (i.e. Soil Moisture [SM]). Another example is at line 194 where the notation "soil moisture" should be edited to "SM".

All the acronyms will be checked and explained as follows: Full Name (FN). For example: European Space Agency (ESA).

**RC19–L130:** Maybe the Authors can mention that T is the *"characteristic time length"*.

As well pointed out, the definition of the T value is not written in the manuscript. It will be added in the revised version.

**RC20-L118:** The term *"estimation"* before images should be removed.

The related sentence will be edited.

---

## Author Comment (AC2)

**The main concerns are listed as follows:**

The main aim of the paper is not clear. Several times it is stated that the aim of the paper is to explore the possibility of substituting the source of the NDVI data (Sentinel-3 instead of Sentinel-2), but this actually occupies a very small portion of the manuscript. The authors conclude that the performances are comparable, so that go in detail on comparisons between the S2MP output against other products and in-situ data. Hence, is this a paper aiming at exploring the use of Sentinel-3 instead of -2 in the processing chain or is it a validation study of the S2MP retrieval (hence an extension of Bazzi et al., 2019)? The paper should be better organized under the perspective of highlighting the main aim of this research.

We truly thank the reviewer for his/her pertinent comments on the manuscript. We understand that the exact goal was not very clear. There is a thoughtful evaluation of different soil moisture products but the goal of the study is not just an evaluation. The goal of the manuscript is to study how to produce 1-km soil moisture maps over large regions using the synergies of different Sentinel satellites. For that, it is needed to (i) extend the $S^2MP$ to other land cover types than croplands and (ii) evaluate the use of S3 instead of S2. The $S^2MP$ was originally designed to work at very high resolution (~10 m). Only afterwards, the relative performances of these maps are evaluated against other state-of-the-art datasets. In a revised version of the manuscript we will clarify the goal and improve the structure. In particular, taking into account this and the other reviewer comments, after comparing S2 and S3 NDVI and the S1+S2 and S1+S3 soil moisture estimations, the rest of the paper will present the S1+S3 maps.

The known issue of the S2MP-derived product consisting in unreliable SM estimates associated with NDVI > 0.7 (Bazzi et al., 2019). Is there any benefit in this sense by using Sentinel-3? Is the issue attenuated working at 1 km?

When the vegetation is well-developed, the estimation of the soil moisture becomes less reliable due to the penetration capabilities of the C-band SAR data in some developed vegetation covers. This limitation has been assessed by several studies showing that for NDVI values beyond 0.7 (NDVI > 0.7), the soil moisture estimation becomes less accurate. This threshold is only a good descriptor of the developed vegetation and is independent from the algorithm used to estimate soil moisture. In other words, when the vegetation is well-developed, the C-band SAR signal will fail to penetrate the vegetation cover and the soil moisture estimation will be less accurate whether the NDVI is calculated from S2 or S3.
Of course, it is more common to have NDVI lower than 0.7 with S3 data because the NDVI value for a 1 km x 1 km pixel includes land parcels with NDVI lower and higher than 0.7 and corresponds to a weighted mean of all NDVI in the pixel of 1 km resolution. But this does not change anything with respect to the fact that S1 can saturate locally. This will be an issue to extend the maps to forest dominated regions but it is not a significant problem for crops and herbaceous areas. In addition, NDVI is a predictor used as input to the neural network that helps to take into account the effect of the vegetation, but the main predictor to estimate the soil moisture is the microwave backscattering.

**Additional minor issues are listed in the following:**

Independently of the SM normalization expressed in eq. (1), I believe that the comparison with CoperSWI should be carried out by calculating the SWI (with same T value) for the S2MP SM as well.

This study is dedicated to surface soil moisture retrieval. In contrast to the so-called change-detection approaches, the $S^2MP$ algorithm produces SM maps in physical units. Studying the soil moisture dynamics in deeper layers using different T values is out of the scope of the study. Therefore, using well-validated and published results, we took the T values that represent the surface layer before transforming the index SWI into physical units. Comparing the two datasets without scaling the indices will also prevent the inter-comparisons with other datasets since most of them are provided in volumetric units.

Potential impacts of a lower coverage of Sentinel-derived observations outside Europe should be discussed.

In contrast to change-detection approaches, which are by construction time-series based approaches, the $S^2MP$ algorithm performs instantaneous retrievals, without using previous values. Therefore, the revisit frequency does not affect the retrievals. We thank the referee for this comment, we will add a short discussion on this topic in the discussion section of a revised version of the manuscript. Regarding the final soil moisture time series, of course the temporal frequency of S1+S2 or S1+S3 will be limiting for some applications but not for long-term soil moisture monitoring at high resolution.

The purpose of the Sentinel-2 VS Sentinel-3 NDVI comparison at 1 km is not clear to me, since Sentinel-2 data has been processed at higher resolution within the S2MP algorithm.

The $S^2MP$ algorithm was designed to work at very high resolution and the S1 data is actually aggregated at high resolution within the 1 $km^2$ pixels only for croplands and herbaceous vegetation areas. When using S2 to estimate NDVI, only those areas are used while in the S3 NDVI all areas within the 1 $km^2$ pixels contribute to the NDVI estimation. It was then necessary to evaluate the potential effect of this mismatch on the way S2 and S3 are used. Therefore, we first compared the NDVI from S2 aggregated for croplands and herbaceous areas to that obtained with S3. One must bear in mind that NDVI is just a predictor used as input to the neural network, but it is not the main predictor (that with the highest weight). Hence, the impact of this possible mismatch is limited. However, it was still needed to evaluate the impact of replacing S2 by S3 in the final result, and this was done by comparing S1+S2 and S1+S3 retrievals to in situ measurements and other soil moisture products. We will clarify the interest of those two comparisons (NDVI first, and final SM estimates afterwards) in a corrected version of the manuscript.

In addition, it is noteworthy that even if the temporal revisit of the SM estimates derived from $S^2MP_{S1S2}$ and $S^2MP_{S1S3}$ is equivalent, the higher temporal revisit of S3 allows to retrieve more optical images without cloud conditions than S2. This results in a better estimation of NDVI.

Lines 273-276. Are the differences attributable to NDVI only? The aggregation to 1 km of Sentinel-1 VV backscattering and of the incidence angle in the S2MP adapted to Sentinel-3 has no impacts?

Of course, there are two reasons that explain the differences between the S1+S2 and S1+S3 soil moisture maps.
(i) Lower the correlation between S2 and S3 NDVI is, higher the difference in soil moisture estimation is. This is due to the fact that S3 NDVI at 1-km resolution integrates NDVI of all types of land parcels within the pixel while S2 NDVI at 1-km resolution only integrates parcels corresponding to croplands and herbaceous vegetation.
(ii) Backscattering coefficients at 10-m resolution are aggregated at 100-m and the neural network is only applied over croplands and herbaceous vegetation for the S1+S2 maps. The SM estimates are then aggregated at 1-km resolution.

In contrast, for the S1+S3 maps, backscattering coefficients over croplands and herbaceous vegetation are aggregated from 10-m to 100-m resolution, and then to 1-km resolution only over croplands and herbaceous vegetation. Finally, the SM estimation is performed at 1-km resolution.

These differences in aggregation (S1 aggregation + SM aggregation for $S^2MP_{S1S2}$, S1 double aggregation for the $S^2MP_{S1S3}$) result in differences between the S1+S2 and S1+S3 soil moisture maps.
These two sources of differences will be reminded in the discussion section.

The higher correlation between in-situ and CR data with respect to HR estimates. Can it be due to the higher temporal resolution of the CR data sets?

The temporal sampling of the CR datasets (SMAP, SMOS, ESA CCI) is roughly 5 times higher than those of the HR datasets ($S^2MP_{S1S2}$, $S^2MP_{S1S3}$, CoperSSM, SMAPS1) and might actually affect the evaluation results with respect to in-situ measurements. Hence, the evaluation of the CR time series has been performed a second time by taking into account only 1 observation out of 5. However, the performances are not affected significantly (see Tables below). In a revised version of the manuscript, we will add a sentence on this check in the results section.

Here are the results without reducing the temporal revisit:

| Products | $R$ | $R^a$ | $Bias$ | $STDD$ |
|---|---|---|---|---|
| Sentinel-only high resolution data | | | | |
| $S^2MP_{S1S2}$ | 0.59 | 0.36 | -0.06 | 0.05 |
| $S^2MP_{S1S3}$ | 0.56 | 0.37 | -0.06 | 0.06 |
| $CoperSSM$ | 0.53 | 0.18 | 0.04 | 0.08 |
| Merged high resolution data | | | | |
| $CoperSWI$ | 0.74 | 0.46 | 0.05 | 0.05 |
| $SMAPS1$ | 0.64 | 0.35 | -0.03 | 0.06 |
| Coarse resolution data | | | | |
| $SMAPL3$ | 0.76 | 0.58 | -0.04 | 0.05 |
| $SMAPL3E$ | 0.77 | 0.59 | -0.04 | 0.05 |
| $SMOSL3$ | 0.67 | 0.47 | -0.03 | 0.07 |
| $SMOSNRT$ | 0.68 | 0.46 | -0.03 | 0.05 |
| $CCISM$ | 0.71 | 0.50 | 0.03 | 0.05 |
| High resolution data aggregated to coarse resolution | | | | |
| $S^2MP^*_{S1S2}$ | 0.58 | 0.38 | -0.06 | 0.06 |
| $S^2MP^*_{S1S3}$ | 0.56 | 0.38 | -0.05 | 0.06 |
| $CoperSSM^*$ | 0.53 | 0.20 | 0.05 | 0.07 |
| $CoperSWI^*$ | 0.73 | 0.47 | 0.05 | 0.05 |
| $SMAPS1^*$ | 0.79 | 0.44 | -0.02 | 0.04 |

Here are the results after reducing the temporal revisit of the coarse resolution datasets:

**Table 4.** Evaluation of the HR and CR SM time series against in-situ measurements in terms of Pearson correlation ($R,R^a$), bias (remotely sensed minus ground based SM in [$m^3\,m^{-3}$]) and standard deviation of the difference ($STDD$ in [$m^3\,m^{-3}$]). The metrics were computed by taking into account the 6 regions of study together and only the median values are shown here. The symbol $^*$ indicates the HR datasets averaged at 25-km resolution. The analysis was performed from January to December 2019.

| Products | $R$ | $R^a$ | $Bias$ | $STDD$ |
|---|---|---|---|---|
| Sentinel-only high resolution data | | | | |
| $S^2MP_{S1S2}$ | 0.59 | 0.36 | -0.06 | 0.05 |
| $S^2MP_{S1S3}$ | 0.56 | 0.37 | -0.06 | 0.06 |
| $CoperSSM$ | 0.53 | 0.18 | 0.04 | 0.08 |
| Merged high resolution data | | | | |
| $CoperSWI$ | 0.74 | 0.46 | 0.05 | 0.05 |
| $SMAPS1$ | 0.64 | 0.35 | -0.03 | 0.06 |
| Coarse resolution data | | | | |
| $SMAPL3$ | 0.81 | 0.55 | -0.05 | 0.05 |
| $SMAPL3E$ | 0.81 | 0.52 | -0.05 | 0.05 |
| $SMOSL3$ | 0.69 | 0.49 | -0.03 | 0.06 |
| $SMOSNRT$ | 0.76 | 0.45 | -0.02 | 0.05 |
| $CCISM$ | 0.73 | 0.48 | 0.03 | 0.05 |
| High resolution data aggregated to coarse resolution | | | | |
| $S^2MP^*_{S1S2}$ | 0.58 | 0.38 | -0.06 | 0.06 |
| $S^2MP^*_{S1S3}$ | 0.56 | 0.38 | -0.05 | 0.06 |
| $CoperSSM^*$ | 0.53 | 0.20 | 0.05 | 0.07 |
| $CoperSWI^*$ | 0.73 | 0.47 | 0.05 | 0.05 |
| $SMAPS1^*$ | 0.79 | 0.44 | -0.02 | 0.04 |

Line 435. Maybe this is due to the fact that the CR component in such data sets is more conclusive than the HR one.
Yes, this is a good remark. Taking into account the results on the HR and CR comparisons with respect to in situ measurements this could actually be the case.

Figure 1. Please add an image explicating the location with respect to the countries.

Please find below a new version of Figure 1. As advised, it has been modified to help the reader to better locate the different regions of studies. Locations of in-situ stations have also been added. This new figure will replace the original Figure 1 into the revised manuscript.

[Figure]

**Figure 1. Panel A:** Global locations of the 6 regions of study. **Panel B:** Copernicus land cover maps of the 6 regions of study aggregated at 1-km spatial resolution. Only the dominant land cover type within a 1-km$^2$ pixel is shown. For instance, a pixel characterised as forests can contain 27% of forests, 26% of croplands, 24% of herbaceous vegetation and 23% of shrublands, or 90% of forests and 10% of herbaceous vegetation. The in-situ stations are shown as black dots. One black dot can correspond to several sensors since some of them have the same coordinates. **Panel C:** Proportion of croplands and herbaceous vegetation within each 1-km$^2$ pixel for the 6 regions of study. The proportion is expressed as a percentage ranging from 0 to 1. Pixels with no cropland or herbaceous vegetation at all are shown as white areas.

Figure 4. The low spatial variability of the shown indices makes the figure not so informative.

Please find a new version of Figure 4 where the colorbar minimums and maximums have been adapted to better match the actual dynamical range of the metrics (R, bias and STDD) and to better show the spatial variations. This new figure will replace the original Figure 4 in the revised manuscript.

[Figure]

**Figure 4.** Comparison of $S^2MP_{S1S3}$ with respect to $S^2MP_{S1S2}$ over the regions of study in terms of Pearson correlation ($R$) as well as bias ($S^2MP_{S1S2}$ minus $S^2MP_{S1S3}$) and standard deviation of the difference ($STDD$) in m$^3$ m$^{-3}$. The analysis was performed from January to December 2019.

---

## Author Comment (AC3)

**General Comments**

The manuscript presents an approach for retrieving the soil moisture (SM) at high resolution (1 km) by using fine resolution observations of Sentinel-1 (Synthetic Aperture Radar Imaging based backscatter coefficient), and Sentinel-2 & 3 (Optical Imaging based NDVI). For high-resolution soil moisture retrievals, the authors adopted the S2MP (Sentinel-1/Sentinel-2 derived Soil Moisture Product) algorithm developed by (El Hajj et al., (2017). The authors used the same approach/ methodology as used for S2MP (neural network with a combination of the Water Cloud Model). This study aims to extend the S2MP from croplands (cereals and grasslands) to herbaceous vegetation types and to explore the use of NDVI derived from Sentinel-3 (S3) instead of Sentinel-2 (S2). The authors provided a comparative analysis of high-resolution soil moisture retrieved through a combination of S1+S2, S1+S3, available soil moisture products of Copernicus Global Land Service (CGLS) SM and Soil Water Index (SWI) and SMAP+S1. For the evaluation of the soil moisture product, the authors used in-situ soil moisture measurements.

Though the topic of the research is important and interesting, I feel there is not much novelty in this research work on high-resolution soil moisture retrieval. The authors adopted a developed approach with only a change of new observations for NDVI (used Sentinel-3 in place of Sentinel-2) without any other improvement/modification. Besides, I feel the authors fail to properly justify why there is a need to use the optical remote sensing-based NDVI to retrieve soil moisture at high-resolution (1 km), which is affected by cloud cover conditions. Since the SMAP-Sentinel products are capable of providing soil moisture at 1 km in all weather conditions, the authors need to identify/justify the adequate research gap to make a novel research statement. On the other hand, extending land cover conditions from "croplands" to "herbaceous vegetation" and using NDVI derived using "Sentinel-3" observations in place of "Sentinel-2) is not a novel research contribution.

Other than scientific fairness, the manuscript structure is poor and needs much improvement. A well-structured "Methodology" section is also missing. A lot of information is redundant and repeated many times in the manuscript. The authors provided a lot of the details on "Datasets" which are well documented in the scientific literature but fail to provide a clear "Methodology" of how the dataset and algorithms are being used. Besides, the manuscript is poorly organized and lacks coherence. The connection in different sections is missing which creates difficulty in understanding the manuscript.

The specific Major/Minor/Editorial (syntax error) issues are listed below.

We truly thank the reviewer for his/her pertinent comments on the manuscript. We regret that the novelty of the study, the global goal and the methodology seem to have been inadequately presented. The goal of the manuscript is to study how to produce 1-km soil moisture maps over large regions using the synergies of different Sentinel satellites. For that, it is needed to (i) extend the $S^2MP$ to other land cover types than croplands and (ii) evaluate the use of S3 instead of S2 NDVI. The $S^2MP$ was originally designed to work at very high resolution (~10 m). Only afterwards, the relative performances of these maps are evaluated against other state-of-the-art datasets. In a revised version of the manuscript we will clarify the goal through the abstract and introduction.

In addition, the descriptions of the Sentinel data and S$^2$MP algorithm will be entirely revised and the structures of the sections data and method will be improved. The section data will be revised to only present the description of the Sentinel missions, the Sentinel data used for the production of the S$^2$MP SM maps and the description of the other well-documented datasets will be shortened. In contrast, the description of S$^2$MP will be moved to section "Methodology". Repetitions as well as not essential descriptions from these sections will be removed in a concerted version.

The SMAP+S1 dataset is certainly a very interesting one and a nice replacement of the originally planned SMAP active / passive dataset. The need for a global product with a resolution of around 1 km is already expressed by several international committees such as GCOS and CEOS and it will increase in the next years. This is certainly an open research topic that requires developing and testing different approaches using different sensors and different methodologies.

**Major comment:**

Notably, the reported research is just an adaptation of the previous approach (El Hajj et al., (2017)) without any other improvement/modification. Since the only changes in the study are "Herbaceous vegetation" land cover in addition to "cropland" and the use of Sentinel-3, I feel there is not much novelty in this research work. The validation of high-resolution soil moisture retrievals on "Herbaceous vegetation" using in-situ measurement is not properly investigated. I can find some correlation comparison in Table 2, but the bias and standard deviation of the difference is not presented for the "Herbaceous vegetation" in Table 3. Besides, the discussion on the different statistics (R2, bias, and STDD) missing in the context of their significance (i.e., are these statistics fulfill the accuracy requirement/goal).

Table 3 presents the performances of the different SM datasets against in-situ measurements in terms of correlation (R), bias, and standard deviation of the difference (STDD). While being a significant extension in the type of regions where the S$^2$MP approach has been tested and validated with respect to El Hajj et al. and Bazzi et al., this study is limited to 6 regions of $10^4$ km$^2$ and then a small number of in-situ stations are available for the evaluation. This is the reason why the metrics (R, bias and STDD) cannot be computed separately for each type of land cover.

However Table 2 does present the comparison between the S$^2$MP SM maps and those from the other high resolution datasets in terms of R over pixels that include a fraction of croplands or herbaceous vegetation. Please find below a new version of Table 2 that presents the bias and STDD in addition to R over pixels dominated by croplands or herbaceous vegetation. This new table can replace Table 2 in a revised version of the manuscript. The addition of these additional metrics in this table do not change the conclusions discussed in the paper but we fully agree with the reviewer that they can be shown for completeness.

Regarding their significance, one must bear in mind that they are just datasets intercomparisons and not absolute evaluation/validation results with respect to a

"ground truth" (which by the say does not exist: calibration and uncertainties of in situ measurements, depth and spatial representativeness issues…). For instance, regarding the STDD (or unbiased-RMS as some colleagues call it now), the relative values obtained for HR maps for herbaceous regions are in between 0.04-0.06 m3/m3, so, similar but a bit larger than the ideal overall performances of remote sensing and model-based products usually quoted in validation documents. But they are similar values with respect to those obtained for croplands.

**Table 2.** Comparison of $S^2MP_{S1S3}$ against $CoperSSM$, $CoperSWI$ and $SMAPS1$ , in terms of $R$, bias and $STDD$, over 1-km$^2$ pixels where croplands or herbaceous vegetation are the dominant land cover classes, respectively. The metrics are also derived according to the degree of coverage of the land cover. One set of metrics is computed considering only pixels covered by less than 75% of croplands or herbaceous vegetation. Another set of metrics is computed considering only pixels covered by at least 75% of croplands or herbaceous vegetation. The analysis was performed from January to December 2019.

| Regions | Products | Croplands | | | | | | Herbaceous vegetation | | | | | |
|---|---|---|---|---|---|---|---|---|---|---|---|---|---|
| | | < 75% | | | ≥ 75% | | | < 75% | | | ≥ 75% | | |
| | | $R$ | $Bias$ | $STDD$ | $R$ | $Bias$ | $STDD$ | $R$ | $Bias$ | $STDD$ | $R$ | $Bias$ | $STDD$ |
| Spain | $CoperSSM$ | 0.63 | 0.01 | 0.06 | 0.72 | 0.01 | 0.06 | 0.67 | 0.06 | 0.06 | 0.69 | 0.07 | 0.06 |
| | $CoperSWI$ | 0.61 | 0.01 | 0.06 | 0.68 | 0.01 | 0.06 | 0.65 | 0.04 | 0.06 | 0.70 | 0.06 | 0.06 |
| | $SMAPS1$ | 0.54 | -0.01 | 0.07 | 0.65 | -0.01 | 0.06 | 0.64 | 0.06 | 0.08 | 0.71 | 0.09 | 0.08 |
| Tunisia | $CoperSSM$ | 0.37 | -0.02 | 0.06 | 0.55 | -0.01 | 0.05 | 0.32 | -0.01 | 0.06 | 0.31 | -0.01 | 0.06 |
| | $CoperSWI$ | 0.37 | 0 | 0.05 | 0.45 | -0.01 | 0.05 | 0.36 | 0.01 | 0.05 | 0.34 | 0.01 | 0.05 |
| | $SMAPS1$ | 0.38 | -0.03 | 0.07 | 0.51 | -0.03 | 0.06 | 0.32 | -0.02 | 0.07 | 0.27 | -0.01 | 0.07 |
| Southwest of France | $CoperSSM$ | 0.56 | 0.03 | 0.06 | 0.71 | 0.03 | 0.06 | 0.69 | 0.07 | 0.06 | 0.62 | 0.09 | 0.06 |
| | $CoperSWI$ | 0.48 | 0.06 | 0.06 | 0.59 | 0.05 | 0.06 | 0.55 | 0.09 | 0.06 | 0.58 | 0.09 | 0.06 |
| | $SMAPS1$ | 0.28 | 0.01 | 0.09 | 0.48 | -0.01 | 0.07 | 0.36 | 0.09 | 0.10 | 0.34 | 0.11 | 0.10 |
| Southeast of France | $CoperSSM$ | 0.63 | -0.04 | 0.04 | 0.75 | -0.05 | 0.03 | 0.44 | 0.02 | 0.05 | 0.72 | 0.06 | 0.02 |
| | $CoperSWI$ | 0.47 | -0.03 | 0.04 | 0.56 | -0.03 | 0.04 | 0.48 | 0.06 | 0.05 | - | - | - |
| | $SMAPS1$ | 0.45 | 0.02 | 0.07 | 0.45 | 0.04 | 0.08 | 0.49 | 0.04 | 0.05 | - | - | - |
| Australia | $CoperSSM$ | - | - | - | - | - | - | - | - | - | - | - | - |
| | $CoperSWI$ | - | - | - | - | - | - | - | - | - | - | - | - |
| | $SMAPS1$ | 0.56 | 0.05 | 0.06 | 0.59 | 0.05 | 0.06 | 0.55 | 0.05 | 0.06 | 0.60 | 0.06 | 0.05 |
| North America | $CoperSSM$ | - | - | - | - | - | - | - | - | - | - | - | - |
| | $CoperSWI$ | - | - | - | - | - | - | - | - | - | - | - | - |
| | $SMAPS1$ | 0.68 | 0.07 | 0.06 | 0.71 | 0.08 | 0.06 | 0.58 | 0.06 | 0.06 | 0.53 | 0.09 | 0.06 |

My other concern regarding validation is "why did the authors not calculate the "RMSE" and "unbiased-RMSE" error matrices which are the most important/critical statistics being used in satellite soil moisture product validations?

We did. The standard deviation of the difference has been used all along the study and this metric is identical to the unbiased-RMSE. Since the bias has also been computed independently, we think the computation of the RMSE will not add value to the study. In addition, those are just some metrics among others. For instance, it is relatively easy to have pretty good STDD (unbiased RMS) with a time series that is flat but close to the mean of the real time series but without really capturing the temporal dynamics. Therefore, other metrics such as different types of correlation coefficient computations are as important as any STD or RMS.

I feel that the proposed approach suffers from the cloud cover situation due to the dependency on optical remote sensing-based NDVI observations. Since the approach of this study mostly depends upon the NDVI in addition to the SAR backscatter, the

approach might fail during cloud cover conditions. Though the authors used a gap-filling linear interpolation approach to obtain two cloud-free NDVI images per month (1$^{st}$ and 15th of each month), this approach still has limitations during long (> 10-15 days) rainy seasons or cloud cover conditions. Besides, it's worthful to use only two NDVI images (15 days apart) in the month to retrieve daily high-resolution soil moisture where NDVI is an important component of the algorithm?

Of course, clouds are something that has to be taken into account when using optical data as input. The advantage of using S3 is that due to its high revisit it is not likely to have not a single cloud free day in 15 days except in some tropical regions with dense vegetation forest, which currently are out of the scope of the S$^{2}$MP algorithm application. Otherwise, one must bear in mind that the NDVI is just a predictor used as input to the neural network, but it is not the main predictor (that with the highest weight) to estimate soil moisture. In addition, the higher temporal revisit of S3 compared to S2 allows the instruments onboard to acquire optical images without cloud or rainy conditions more frequently. Thus, only several cloud-free optical images are required to compute robust NDVI estimates. And NDVI variation in a 15 days timescale is slow enough to have the flexibility of using this auxiliary predictor variable with this timescale and therefore minimizing the risk of having gaps due to clouds. Finally, there must be a misunderstanding, because not only two images per month are used. In the construction of those two images all S3 images have been used. We will clarify this if we are asked to send a revised version of the manuscript.

Although the authors have presented the details on the use of optical remote sensing (which is susceptible to errors associated and large data gaps due to the clouds, and atmospheric effects) with microwave remote sensing (active/passive) for high resolution soil moisture retrievals, a proper justification or criticism is missing between the synergistic use of purely microwave remote sensing-based approach like SMAPSentinel active-passive approach. I suggest the authors should provide justifications in this regard. My concern is "if SMAP-Sentinel has the capability to provide 1-km soil moisture product using Sentinel-1 SAR observations to a global extent then what is the value addition with the proposed approach, which also uses Sentinel-1 SAR observations to provide 1-km soil moisture retrievals which are limited only to the study regions? Is the performance of the proposed approach better than the SMAPSentinel to provide high-resolution soil moisture? If yes then provide adequate analysis and proper comparison. If not, then justify why this study is important.

As already mentioned, the SMAP+S1 dataset is certainly a very interesting one and a nice replacement of the originally planned SMAP active / passive dataset. The need for a global product with a resolution of around 1 km is already expressed by several international committees such as GCOS and CEOS and it will increase in the next years. This is certainly an open research topic that requires developing and testing different approaches using different sensors and different methodologies.
For instance, downscaling by data merging is not an exact science and part of the high spatial frequencies in the merged dataset are noise. The fact that, when compared to in situ measurements, the performances of SMAP+S1 increase significantly when the downscaled data are aggregated back at 25 km resolution is an example. We do not

share the point of view that no more research in the topic is needed because there is a SMAP+S1 product. Hundreds of works in the remote sensing of soil moisture litterature have already shown that active or passive, single resolution/sensor or merging of different revolution datasets, using optical data such as NDVI as auxiliary data (as SMAP does because it cannot retrieve simultaneously soil moisture and optical depth) or only microwaves, using physical, statistical or change detection approaches... sometimes perform better in some places and some periods and in other regions/times it is another dataset that perform the best. Or simply, they differ significantly as in the Equatorial and Boreal regions where third party data, in particular in-situ measurements, is not available to determine which one can be closer to the reality.

The authors did not show any spatial pattern of the high-resolution soil moisture retrievals. I suggest the authors show a few spatial maps (i.e., dry, wet, and moderate soil moisture conditions) of the retrieved soil moisture using the proposed approach and its comparison to SMAP-Sentinel products. Since both the products are based on the Sentinel-1 observations, there will be a similar areal coverage in both the products and will help to understand the spatial distribution of high-resolution soil moisture and the reasoning behind the error difference.

We do think that comparing to SMAP+S1 is interesting and this is the reason that we used it in the comparison. However, we disagree with the idea that it should be the main reference. The main reference is the Copernicus surface soil moisture dataset because it is S1 based without merging with a very different resolution sensor, which introduces additional uncertainties in the comparison. However, we do agree that it is interesting to compare the datasets in shorter time periods than one year and to show maps in different time periods. For instance, please find below a figure showing soil moisture maps averaged over the period of study (from January to December 2019) for each dataset and region.

[Figure]

**Figure 4.** SM mean of $CoperSSM$, $CoperSWI$, SMAPS1 and $S^2MP_{S1S3}$ over the 6 regions of study in 2019.

The mean maps are quite similar between the Copernicus SM and SWI (CoperSSM and CoperSWI) datasets for all the 4 regions in Europe.

In Spain, $S^2MP$ is quite similar to the Copernicus and SMAP+S1 maps. However, SMAP+S1 is wetter in the southwest of this region and $S^2MP$ is dryer.

In Tunisia, SMAP+S1 is drier than the 3 other datasets. $S^2MP$ shows wetter SM estimates than the other datasets over pixels close to river basins.

In the southwest of France, SMAP+S1 shows wetter SM estimates than the Copernicus datasets over forest areas while $S^2MP$ shows dryer SM estimates.

In the southeast of France, SMAP+S1 is clearly wetter than the Copernicus datasets. Over the regions in Australia, in particular over herbaceous vegetation, and America, $S^2MP$ is dryer than SMAP+S1.

The introduction needs much improvement. Firstly, the manuscript needs to critically discuss why this study is important. What is the novel research statement/objective of this study? Secondly, the introduction needs details for an international context. How do the findings of this study inform or build upon the wide range of international research that has been carried out in high-resolution soil moisture retrievals? What does this research contribute? Since ANN-based retrievals are limited only to the study regions, what information from this study will be relevant to international researchers outside of the specific six regions location investigated?

Retrieving soil moisture is currently done globally at low spatial resolution (40 km) or over smaller regions at high (~1 km) or very high spatial resolution (10-100 m) as the

original S²MP algorithm does over croplands. The goal of this manuscript is to study how the S²MP approach could be used to produce 1-km soil moisture maps over large regions using the synergies of different Sentinel satellites. Of course, the first step is to be able to use it over larger regions and to extend S²MP to other land cover types than croplands. In addition, since for soil moisture mapping at large scale, 1 km is a high spatial resolution, using S3 instead of S2 becomes possible. In spite of the lower spatial resolution of S3, its higher revisit frequency with respect to S2 makes possible to reduce uncertainties in the NDVI estimations used by S²MP introduced by clouds and gaps in the optical time series.

In this context, we understand that comparing the S1+S2 vs S1+S3 results and continuing showing the performances of S1+S2 against those of other state-of-the-art datasets was a bad choice. In a revised version of the manuscript we will focus on the results for S1+S3 in addition to clarify the goal in the abstract and introduction as well as improve the structure and content of the data and method sections.

Finally, this manuscript is also an evaluation of the S²MP approach over regions with different conditions with respect to those in El Hajj et al. and Bazzi et al. papers.

ANN-based retrievals are not limited to the six regions. There is no local data used for the training but this paper does not present any kind of operational algorithm, it is a research paper, we cannot process the whole globe.

The "Conclusions" section is full of results only. I feel the conclusion should be a take home message for the readers and should be related to the work's problem statement in a concise manner. Please revise this section.

We appreciate the feedback of the reviewer and we will certainly revise this section making it more concise and including perspectives to the summary section.

**Minor comments:**
L4: "agricultural plot scale"- What scale are you talking about? It should be quantitative. Since the proposed method is for 1-km soil moisture, using the term "agricultural plot scale" is not optimistic.

This sentence will be edited to explicitly specify that the agricultural plot scale corresponds to a 10-m resolution.

L9-10: "A target resolution of 1 km also…"- In what way does 1-km spatial resolution allows to explore the use of NDVI derived from Sentinel-3 (S3) instead of S2? Is S2 not having the potential to provide NDVI at 1-km?

We understand that this sentence is not clear. Both S2 and S3 have the potential to provide NDVI at 1-km resolution. This sentence will be edited to avoid potential misunderstandings.

The authors need to revise the section "Section 2. Data". I suggest providing brief details about the well-known datasets. Most of the details look redundant.

Taking into account this comment and those from other reviewers, the section data will be entirely revised to include a better description of the Sentinel data used as input to the $S^2MP$ algorithm and to avoid non-essential information concerning the well-known datasets used in this study.

Figure-1 is missing the details of in-situ soil moisture measurement locations.

Please find below a new version of Figure 1. As advised, it has been modified to help the reader to better locate the different regions of studies. Locations of in-situ stations have also been added. This new figure will replace the original Figure 1 into the revised manuscript.

[Figure]

**Figure 1. Panel A:** Global locations of the 6 regions of study. **Panel B:** Copernicus land cover maps of the 6 regions of study aggregated at 1-km spatial resolution. Only the dominant land cover type within a 1-km$^2$ pixel is shown. For instance, a pixel characterised as forests can contain 27% of forests, 26% of croplands, 24% of herbaceous vegetation and 23% of shrublands, or 90% of forests and 10% of herbaceous vegetation. The in-situ stations are shown as black dots. One black dot can correspond to several sensors since some of them have the same coordinates. **Panel C:** Proportion of croplands and herbaceous vegetation within each 1-km$^2$ pixel for the 6 regions of study. The proportion is expressed as a percentage ranging from 0 to 1. Pixels with no cropland or herbaceous vegetation at all are shown as white areas.

Table-1: In North America, USCRN locations consist of only two measurement locations. Are two locations optimal to represent the spatial distribution with 1-km

grid cells? Past studies show that at least 3 locations are required to up-scale the soil moisture within a 1-km grid-cell.

The way in situ measurements have been used along the study will be clarified in the section method in the revised manuscript. We confirm that more than 3 in-situ stations have been used to scale the Copernicus data for each region. In the new version of the manuscript, we will distinguish the in-situ measurements used for the scaling from those used for the evaluation of the remotely sensed time series. Indeed, several in-situ stations have been discarded from the evaluation because some metrics (R, bias, STDD) computed between the in-situ and remotely sensed time series were not significant (P value below 5% - Interval of confidence of 95%). This distinction is actually not present in the manuscript and only in-situ measurements used for the evaluation are shown. Table 1 will be updated as follows.

**Table 1.** In-situ measurements that were used in this study. The depths are quoted as two numbers: the first one is the upper depth, and the second one is the lower depth of the sensor. Both numbers are equal when the sensor is placed horizontally. The fourth column gives the number of sensors that provide SM measurements in 2019. These measurements were used to convert the relative indices from $CoperSSM$ and $CoperSWI$ into SM estimates with volumetric units (m$^3$ m$^{-3}$, Section 3.2). The number in parenthesis corresponds to the number of in-situ locations where the evaluations of the remotely sensed data were significant (P-value below 5%, Section 3.3).

| Measurements | Location | Depth (m) | Sensors | Reference |
|---|---|---|---|---|
| REMEDHUS | Spain | 0–0.05 | 19 (12) | Gonzalez-Zamora et al. (2018) |
| SMOSMANIA | Southwest of France | 0.05–0.05 | 4 (3) | Calvet et al. (2007) |
| SMOSMANIA | Southeast of France | 0.05–0.05 | 5 (0) | Calvet et al. (2007) |
| OZNET | Australia | 0–0.05 | 11 (10) | Smith et al. (2012); Young et al. (2008) |
| USCRN | North America | 0.05–0.05 | 2 (1) | Bell et al. (2013) |
| ARM | North America | 0.05–0.05 | 24 (13) | Cook (2016, 2018) |
| MERGUELLIL | Tunisia | 0–0.05 | 5 (2) | Gorrab et al. (2015) |

This was the opportunity to update the in-situ measurements form the ISMN and to include additional sites to the evaluations.

**Editorial comments:**

Authors should be consistent with either "soil moisture estimates" or "soil moisture retrievals" - sometimes authors used "soil moisture dataset" – the terminology used should be consistent throughout the manuscript.

As advised, the terminology will be revised (soil moisture estimates) and will be consistent throughout the manuscript.

L3: What are other purposes?

Meteorology, climate monitoring and climate impact assessment at regional impact, landslide predictions…. we cannot cite all applications of soil moisture in the first introductory sentence of an abstract. Alternatively we could suppress it.

L3: The term "For instance" is not appropriate here.

This term will be removed.

L3-6: "For instance… as inputs to a neural network trained with Water Cloud Model simulations"- the statement is not clear. What is meant by "inputs to a neural network trained with Water Cloud Model simulations"?

This sentence will be edited. To estimate soil moisture, the $S^2$MP algorithm uses a neural network that takes data derived from S1 radar signal (backscattering coefficients) and S2 optical images (NDVI) as inputs. The neural network was trained using a synthetic database gathering (i) SAR C-band backscatter coefficients in the VV polarization (ii) incidence angles (from 20 to 45 degrees), and (iii) NDVI as inputs and soil moisture as target. This synthetic database was built using a Water Cloud Model combined with an Integral Equation Model that was specially modified and optimized for this application. This will be explained in detail in the section method in the revised manuscript.

L6: "However, for many applications…" – Why the use of "However"? Is this statement contradicting statement with the previous one?

This sentence will be edited.

L6 "future climate impact assessment"- why suddenly climate change?

Because there is a need for global 1 km soil moisture data expressed by CEOS, GCOS and ESA CCI users.

L6-8: statement is very long and difficult to understand.

"However, for many applications, including future climate impact assessment at regional level, a resolution of 1 km is already a significant improvement with respect to most of the publicly available SM data sets, which have resolutions of about 25 km" means that for many applications estimating SM at agricultural plot level (previous sentence) is not needed and 1 km will be considered as a real step forward with respect to most available datasets. Even the climate community is already looking forward to this resolution.

L10-11: "…Europe and other regions of the globe, for which S1 coverage is poorer."-revise the statement.

This mention to the coverage can be removed from the abstract because nothing is mentioned first regarding the coverage in Europe to say afterwards that outside Europe is poorer. That's for pointing this out.

L15-16: "…maps were compared to each other and to those of the 1-km resolution Copernicus Global Land Service (CGLS) SM and Soil Water Index (SWI) data sets as well as to the SMAP+S1 product" – this statement has no meaning. Revise it.

We mean that maps were compared to each other. In addition, both were compared to the 1-km resolution Copernicus Global Land Service (CGLS) SM and Soil Water Index (SWI) and the SMAP+S1 datasets".

L25: change "data sets" to "datasets"

The term "data sets" will be changed to "datasets" all along the manuscript.

L25: "HR data sets were also compared …" What high-resolution dataset refers here please specify for clarity.

In this sentence, the HR datasets will be named for a better clarity: $S^2MP_{S1S2}$, $S^2MP_{S1S3}$, CoperSSM, CoperSWI, SMAPS1.

L49: change "data sets" to "dataset" or delete it.

The term "data set" will be changed to "dataset".

L64: change "in situ data" to "in-situ measurements" – correct throughout the manuscript

The term "in-situ measurements" will be used all along the manuscript.

Section "4.3.1 Absolute values" What absolute values refer here : This heading is not complete/and doesn't have a clear meaning-please revise.

The title of this subsection will be changed to "Comparison of the order of magnitude".